# The Autophagic Activator GHF-201 Can Alleviate Pathology in a Mouse Model and in Patient Fibroblasts of Type III Glycogenosis

**DOI:** 10.3390/biom14080893

**Published:** 2024-07-24

**Authors:** Kumudesh Mishra, Sahar Sweetat, Saja Baraghithy, Uri Sprecher, Monzer Marisat, Sultan Bastu, Hava Glickstein, Joseph Tam, Hanna Rosenmann, Miguel Weil, Edoardo Malfatti, Or Kakhlon

**Affiliations:** 1Department of Neurology, The Agnes Ginges Center for Human Neurogenetics, Hadassah-Hebrew University Medical Center, Jerusalem 9112001, Israel; kumudeshmishra@gmail.com (K.M.); sahar.sweetat@mail.huji.ac.il (S.S.); rosenman@hadassah.org.il (H.R.); 2Obesity and Metabolism Laboratory, Institute for Drug Research, School of Pharmacy, Faculty of Medicine, The Hebrew University of Jerusalem, Jerusalem 9112001, Israel; saja.baraghithy@mail.huji.ac.il (S.B.); yossi.tam@mail.huji.ac.il (J.T.); 3The Shmunis School of Biomedicine and Cancer Research, The George S. Wise Faculty for Life Sciences, Sagol School of Neurosciences, Tel Aviv University, Tel Aviv 6997801, Israel; uri8sp@gmail.com (U.S.); monzermar@gmail.com (M.M.); miguelw@tauex.tau.ac.il (M.W.); 4Centre de Reference de Maladies Neuromusculaires, UPEC—Paris Est University, IMRB INSERM U955, Team Biology of the Neuromuscular System, Faculty of Medicine, APHP Hopital Henri Mondor, 1 Rue Gustave Eiffel, 94010 Creteil, France; sultan.bastu@inserm.fr (S.B.); edoardo.malfatti@aphp.fr (E.M.); 5Electron Microscopy Unit, The Hebrew University-Hadassah Medical School, Ein Kerem, Jerusalem 9112001, Israel; hava.glickstein@mail.huji.ac.il; 6Faculty of Medicine, Hebrew University of Jerusalem, Jerusalem 9112001, Israel

**Keywords:** glycogen, glycogen storage disease type III, pharmacotherapy

## Abstract

Glycogen storage disease type III (GSDIII) is a hereditary glycogenosis caused by deficiency of the glycogen debranching enzyme (GDE), an enzyme, encoded by *Agl*, enabling glycogen degradation by catalyzing alpha-1,4-oligosaccharide side chain transfer and alpha-1,6-glucose cleavage. GDE deficiency causes accumulation of phosphorylase-limited dextrin, leading to liver disorder followed by fatal myopathy. Here, we tested the capacity of the new autophagosomal activator GHF-201 to alleviate disease burden by clearing pathogenic glycogen surcharge in the GSDIII mouse model *Agl^−/−^*. We used open field, grip strength, and rotarod tests for evaluating GHF-201’s effects on locomotion, a biochemistry panel to quantify hematological biomarkers, indirect calorimetry to quantify in vivo metabolism, transmission electron microscopy to quantify glycogen in muscle, and fibroblast image analysis to determine cellular features affected by GHF-201. GHF-201 was able to improve all locomotion parameters and partially reversed hypoglycemia, hyperlipidemia and liver and muscle malfunction in *Agl^−/−^* mice. Treated mice burnt carbohydrates more efficiently and showed significant improvement of aberrant ultrastructural muscle features. In GSDIII patient fibroblasts, GHF-201 restored mitochondrial membrane polarization and corrected lysosomal swelling. In conclusion, GHF-201 is a viable candidate for treating GSDIII as it recovered a wide range of its pathologies in vivo, in vitro, and ex vivo.

## 1. Introduction

Glycogen is a branched polysaccharide with a molecular weight of nine to ten million Da. The average glycogen molecule contains about 55,000 glucose residues linked by α-1,4 (92%) and α-1,6 (8%) glycosidic bonds [1]. The synthesis of glycogen is primed by glycogenin [2] (even though aberrant forms of glycogen are also formed when glycogenin is suppressed [3]), and catalyzed by glycogen synthase (GYS) and glycogen branching enzyme (GBE). GYS links glucose residues to each other by α-1,4 glycosidic bonds to form linear chains. GBE, in turn, attaches a short linear stretch of α-1,4-linked glucose units from the outer non-reducing end of a growing glycogen chain into an α-1,6 position of a pre-existing chain, thus generating a branch. Repetitive cycles of elongation and branching generate a large (110–290 nm in liver) [4] spherical molecule with a hydrophilic surface and an increased number of reactive termini, facilitating glycogen synthesis and degradation [5,6]. Glycogen degradation, on the other hand, takes place in the cytosol and in lysosomes. In the cytosol, glycogen is degraded by glycogen phosphorylase (GP), which phosphorylytically releases outer glucose residues in the form of glucose-1-phosphate, and by glycogen debranching enzyme (GDE). GP phosphorylates outer glucose residues and releases each one as glucose-1-hosphate. GP can degrade an outer chain only until four residues from the branch point. Then, GDE is required to complete glycogenolysis. GDE catalyzes two reactions: transferring a three-glucosyl glucan to the non-reducing end of another linear strand (α-1,4-glucanotransferase reaction) and hydrolyzing the α-1-6-glycosidic bond remaining on the branch point, releasing glucose (amylo-α-1,6-glucosidase reaction, *n.b.*, glucose, rather than glucose-1-phosphate, is released [4]). In the lysosomes, glycogen degradation is catalyzed by α-glucosidase.

Glycogen is stored primarily in liver and muscle, where it, respectively, serves as a reservoir for regulating blood glucose levels and as a readily mobilizable energy reserve. Therefore, aberrations in glycogen metabolism, or glycogen storage disorders (GSDs), are primarily hepatic or muscular disorders, with two notable exceptions, APBD and LD, specifically involving the central and peripheral nervous systems. The most common disorder of glycogen metabolism is observed in diabetes [7,8], in which abnormal amounts of insulin or abnormal insulin response result in accumulation or depletion of liver glycogen. However, GSDs are normally associated with hereditary deficiencies in enzymes of glycogen and glucose metabolism causing glycogen accumulation (except for GSD type 0, which is a GYS deficiency and is therefore associated with glycogen insufficiency).

In this work, we focused on GSD type III (GSDIII, Cori disease), the third most prevalent muscle GSD (incidence of 1:100,000) after GSD types V and II [9,10,11], which is caused by GDE deficiency leading to accumulation of glycogen due to its reduced degradation. Phosphorylase-limit dextrin (PLD), the type of glycogen which accumulates in GSDIII, is structurally abnormal, containing shorter outer branches [12]. While phosphorylase-limit dextrin is sensitive to diastase digestion and does not form polyglucosan bodies [12], it is still associated with large vacuoles and can cause histological damage to myofibrils [13,14]. GSDIII usually starts as a liver disorder characterized by hepatomegaly, hypoglycemia, hyperlipidemia, and hyperketonemia. These result from the limited ability to breakdown glycogen to glucose, leading to excessive use of lipid oxidation as an alternative energy source with ensuing attenuation of lipid uptake by adipocytes and increase in ketone bodies as byproducts of fatty acid oxidation (FAO). Later on, myopathy pursues [15], with occasional cardiomyopathy [16], liver cirrhosis, and hepatocellular adenoma [17].

GSDIII implicates vacuolar myopathy and myofibrillar damage [14,18], also observed in GSDII [19], which shares muscle glycogen over accumulation with GSDIII (albeit lysosomal in GSDII and mostly cytoplasmic in GSDIII). We, therefore, tested the GSDIII curative capacity of GHF-201, a novel, safe compound capable of reversing glycogen accumulation and enhancing autophagy [20,21]. We show that GHF-201 is capable of improving motility and grip strength in an *Agl^−/−^* mouse model of GSDIII, where GDE, encoded by the murine *Agl* gene, was knocked out. At the same time, GHF-201 did not modify the deficient motor learning capacities of *Agl^−/−^* mice associated with a neurological deficiency not implicated in GSDIII. We further demonstrate that GHF-201 corrected hypoglycemia, hyperlipidemia, and liver and muscle damage, as well as carbohydrate over fat fuel preference. Combined with reduction of glycogen levels and muscle atrophy at the tissue level, and improvement of lysosomal and mitochondrial phenotypes at the cell level (in GSDIII patients’ fibroblasts), our results strongly support that GHF-201 has a potential therapeutic capacity for treating GSDIII, either as a standalone, or as part of a combination therapy. At the moment, GSDIII has no curative alternative and patients are only managed by dietary restrictions [22]. A clinical trial for GSDIII using the UX053 gene therapy-based replacement of GDE (NCT04990388) has been initiated, but was terminated at phase I. While the gene therapy GDE replacement approach is still valid, it can also be improved by add-on therapy based on a mechanistically complementary approach. In that respect, activation of glycogen degradation by autophagy merits appraisal, especially since rapamycin-mediated activation of autophagy was indeed efficacious in a GSDIII dog model [23].

## 2. Materials and Methods

### 2.1. In Vivo Studies

In vivo studies were conducted on *Agl^−/−^* mice [9] and congenic C57Bl6J mice as littermate wild type controls (*Agl^−/−^*). n = 6–10 mice/arm were used. Twice a week, 50 µL of 250 mg/kg GHF-201 in 10% DMSO was intramuscularly injected to the gastrocnemius in alternate flanks. A 10% DMSO vehicle was used as a vehicle control. Mice were treated from the age of 3 months to the age of 8 months. Animals’ weight was recorded every week. A weight decrease of >10% between biweekly weightings, or >20% from treatment initiation led to removal of the animal from the study. We tested monthly moving distance and cumulative sojourn in the center of an open field, front paw grip strength using a grip strength meter (Columbus Instruments, Columbus, OH, USA), and rotarod performance (latency to fall from a rotating cylinder). The rotating cylinder (UGO Basile S.R.L., Gemonio, Italy) accelerated from 5 to 40 RPM in 99 s. Mice were tested for 3 consecutive trials with a rest period of approximately 10 min between trials.

Blood biochemistry (enzyme activities), triglycerides, and glucose were determined at the Hadassah central laboratory division. Blood was collected from the cheek pouch by submandibular bleeding. Indirect calorimetry and multi-parameter metabolic assessment (metabolic cages) were performed in Promethion Metabolic Phenotyping Systems (Sable Instruments, Las Vegas, NV, USA) as in [20]. Specifically, locomotor activity was assessed in these cages by the number of times an infrared XYZ beam array is disrupted by animal movement, while wheel running was determined by the number of wheel revolutions [24].

### 2.2. Ultrastructural Electron Microscopy Studies

Gastrocnemius muscle samples were fixed, embedded, sectioned, and analyzed by transmission electron microscope as described in [9]. Area percentage of glycogen dense granules in longitudinal muscle sections (10 sections/animal) was quantified in samples from n = 3 vehicle-treated WT, n = 3 vehicle-treated *Agl^−/−^*, and n = 3 GHF-201-treated *Agl^−/−^* mice.

### 2.3. Biochemical Glycogen Quantification

Muscle glycogen was quantified biochemically using Merck’s Glycogen Assay Kit (Cat. # MAK465) according to the manufacturer instructions. In brief, 10–20 mg of muscle tissues was homogenized on ice in 1 mL of 25 mM citric acid, 2.5 mg/mL NaF, pH 4.2. Samples were spun down (14,000× *g*, 5 min, RT) to clear the debris. Then, 10 µL of cleared supernatant was used for the colorimetric determination of glycogen using the kit. A calibration curve of 0–200 µg/mL glycogen (provided with the kit) was used for glycogen quantification. Water was used as a blank control, to which the same reagents were added. This blank reading was subtracted from the colorimetric readings of all samples and calibration standards.

### 2.4. High Content Analysis of GSDIII Patient Fibroblasts

Primary skin fibroblasts from 3 patients were procured from Coriell Institute (Camden, NJ, USA) and expanded in DMEM supplemented with 1% Sodium Pyruvate, 1% Penicillin-Streptomycin-Amphotericin B, 10% heat-inactivated fetal calf serum (Biological industries, Beit HaEmek, Israel), and 1% 100× non-essential amino acids solution (NEAA) at 37 °C with 5% CO_2_. For live imaging, GSDIII and healthy control (HC) skin fibroblasts were seeded at 1400 cells per well and cultured at 37 °C and 5% CO_2_ with or without GHF-201 (50 µM) for 24 h in specialized microscopy-grade 96-well plates (Cellvis P96-1.5H-N). Cells were then washed in HBSS, supplemented for 20 min with 1.6 mM Hoechst 33342, 0.1 mM LysoTracker Deep Red, 0.05 mM TMRE, and 0.4 mM calcein-AM Green (Thermo-Fisher Scientific, Waltham, MA, USA), and washed again with HBSS. Subsequently, images were acquired using an Operetta G1 system at 20× magnification under environmental control (37 °C and 5% CO_2_) conditions. All assay parameters (including the acquisition exposure times, objective, and the analysis parameters) were kept constant for all assay repetitions. Images were analyzed by the Harmony image analysis software version 1.0. Downstream analysis included removal of wells with outlier values using GraphPad Prism (version 10.2.3)’s ROUT method (Q = 10%), feature selection by backward feature elimination, and normalization to HC levels prior to statistical analysis [25]. Multivariate analysis was performed in Python.

### 2.5. Statistical Analysis

Sample sizes were determined based on the effect sizes of key parameters measured (e.g., open field moving distance), as detailed in [26]. These sample sizes were sufficient to provide the acceptable power of at least 80%. Blinding in in vivo studies was obtained by encoding the animals in the different treatment groups so that the experimenters were blinded to treatment allocation. To obtain randomization, litters from mice born a few days apart were pooled and pups of similar weight were divided into vehicle- and GHF-201-treated arms. Specific statistical tests are mentioned in the relevant text and figure legends.

For the indirect calorimetry and metabolic cage analyses (Figure in Section 3.4), mean differences in metabolic parameters between wild type and vehicle-treated *Agl^−/−^* mice, wild type and GHF-201-treated mice and vehicle-treated and GHF-201-treated *Agl^−/−^* mice were calculated during light, dark, and overall (light + dark) periods. One Way ANOVA with repeated measures (and Tukey post hoc validation test) was used to determine the statistical significance of these mean differences, which are the output of repeated measures of ANOVA, for each metabolic parameter during the entire time periods analyzed (light, dark, or overall). We show in the left panels the actual time-dependent trends of the respective metabolic parameter in each treatment arm. In the right panels, we show the mean differences between the group trend averages (n = 4 mice in each group) derived from these time trends. For the food and water intake profiles, since they are cumulative, mean differences were calculated based on subtractions of the time periods end values: at 12 h for the light period, at 24 h minus 12 h for the dark period, and at 24 h for the overall periods. The statistical significance of the mean differences is shown by asterisks above the respective bars. The total energy expenditure (TEE) ANCOVA analysis was provided by the NIDDK Mouse Metabolic Phenotyping Centers (MMPC, www.mmpc.org (accessed on 1 July 2024)) using their Energy Expenditure Analysis page (http://www.mmpc.org/shared/regression.aspx (accessed on 1 July 2024)) and supported by grants DK076169 and DK115255.

## 3. Results

### 3.1. Animal Weight

GSDIII male mice presented with a higher weight throughout the treatment period as compared to WT mice (*p* < 0.0001, Two Way ANOVA with repeated measures). According to [13], GSDIII mice were lighter than WT mice, while in [27], no significant difference between the weights of GSDIII and WT mice was observed, together suggesting that such a difference might be batch-dependent. GHF-201 treatment did not significantly modify weight in GSDIII mice (Figure 1). Animal weighting was discontinued 138 d after treatment initiation due to an unexplained >10% weight drop between two weightings in all animals.

### 3.2. Locomotor Studies

We show that GHF-201 improved open field performance at a relatively advanced stage of the disease (last time point when mice were approximately 8 months old). This improvement was demonstrated as increases in both distance traveled (Figure 2A) and sojourn in the center of the open field (Figure 2B), which indicates a more exploratory behavior associated with decreased anxiety. Sojourn in the center was corrected by GHF-201 to WT values at the age of 8 months. Interestingly, while GHF-201 significantly increased grip strength (Figure 2C) and improved open field parameters, it did not increase the latency to fall from a rotating rod in the rotarod test (Two Way ANOVA with repeated measures, *p* < 0.95, *p* < 0.95, *p* < 0.97 at 10, 47, and 108 days post-treatment initiation) and even decreased it 80 days post-treatment initiation (Two Way ANOVA with repeated measures, *p* < 0.05). Latency to fall was decreased in *Agl*^−/−^ mice as compared to WT at all times post-treatment initiation (Two Way ANOVA with repeated measures, *p* < 0.11 (only a trend), *p* < 0.03, *p* < 0.005, and *p* < 0.02 at 10, 47, 80, and 108 days post-treatment initiation, Figure 2D). Training and motor learning capacity were, respectively, demonstrated by increased latency between runs in each test and by increased overall latency over time. As opposed to untreated and treated *Agl*^−/−^ mice, which showed no training capacity, WT mice manifested training capacity at 80 days post-treatment initiation. This was determined by comparing the average run latencies (shown in different shades of gray, red, and blue in WT, KO/Veh, and KO/GHF-201, respectively) at each time following treatment initiation using Two Way ANOVA with repeated measures (*p* < 0.04). Motor learning capacity of WT mice was also significantly higher than that of untreated or treated *Agl*^−/−^ mice (*p* < 0.0095, R^2^ = 0.2687 for linear correlation of latency v time in WT mice, as compared to (*p* < 0.0436, R^2^ = 0.0732) and (*p* < 0.0691, R^2^ = 0.0523) for untreated and treated *Agl*^−/−^ mice, respectively). Indeed, rotarod measures motor coordination, which is considered a neurological function of the nigrostriatal dopamine system [28,29] not implicated in GSDIII. Motor activity measured by the open field test, on the other hand, is influenced by muscle strength, which was modified by GHF-201 (Figure 2C). There are several works (e.g., [30]) which demonstrate the independence of rotarod and open field results.

Notably, all behavioral tests had to be discontinued following the 4th test post-treatment initiation. The reason for that was that after the 4th test the weight of all mice was decreased by >10% from the last reading suggesting that GHF-201, increases muscle strength-associated mobility, but not survival, in GSDIII mice.

### 3.3. Blood Metabolic Panel

We tested whether GHF-201 is able to correct the hypoglycemia, hypertriglyceridemia, and liver and muscle damage as measured by blood examinations [13,31,32]. Such effects are expected from an agent, such as GHF-201, capable of inducing the catabolism of liver glycogen with an ensuing rise in blood glucose and reduction in compensatory lipid flux to the liver. Our blood metabolic panel results show that GHF-201 is able to correct hypoglycemia and hyperlipidemia and partially restore muscle (creatine kinase) and liver (alanine transferase) damage, all of which characteristically accrue with time in the *Agl*^−/−^ GSDIII mouse model (Figure 3).

### 3.4. In Vivo Metabolic Profile

The effect of GHF-201 on various in vivo metabolic parameters was determined by indirect calorimetry. Fuel preference at the whole animal level is determined by the respiratory exchange ratio (RER, the ratio of CO_2_ produced to O_2_ consumed). Lower RER indicates higher fat burn, while higher RER indicates higher carbohydrate burn. As our results (Figure 4(A1)) show, *Agl^−/−^* mice had lower RER values in the light, dark, and overall (positive mean difference in the black and red bar graphs indicates that WT/Veh levels are higher than KO/Veh and KO/GHF-201, respectively; negative mean difference indicates that they are lower). GHF-201 significantly increased RER in treated v. untreated Agl^−/−^ mice in light, dark, and overall (blue bars; positive values indicate KO/Veh > KO/GHF-201, negative values indicate KO/GHF-201 > KO/Veh). However, as compared to WT mice, GHF-201 treated *Agl^−/−^* mice still had a significantly reduced RER in all periods (red bars). Total energy expenditure (TEE, Figure 4(A3)) was slightly reduced in *Agl^−/−^* mice as compared to WT mice (black bars). Like RER, TEE in *Agl^−/−^* mice was also significantly increased by GHF-201 in light and dark conditions and overall (blue bars), with the most significant increase (*p* < 0.001) in the light condition (*cf*. *p* < 0.05 for dark and overall). Compared to WT, TEE in GHF-201-treated *Agl^−/−^* mice was increased in the light but remained lower than WT in the dark and overall (red bars). Thus, GHF-201 treatment dramatically increased TEE in *Agl^−/−^* mice, where it was slightly reduced as compared to WT. GHF-201 correction of TEE can also be revealed by our ANCOVA analysis (Figure 4(A2),(A4)), performed on light and overall periods, where the TEE effect was the most pronounced, which also accounts for the dependence of TEE on body weight. According to the ANCOVA analysis, GHF-201 corrected weight-adjusted TEE to WT levels: the difference between WT and *Agl^−/−^* in light and overall conditions became insignificant when *Agl^−/−^* mice were treated with GHF-201. Importantly, similar to TEE correction to WT level by GHF-201, a significant difference in TEE between vehicle- and GHF-201-treated *Agl^−/−^* mice is also revealed by ANCOVA only under the light condition. As opposed to RER and TEE, fat oxidation (Figure 4(A5)) in *Agl^−/−^* mice was higher than in WT mice in all periods (black bars). GHF-201 increased fat oxidation in *Agl^−/−^* mice only in the light (blue bar) and maintained the higher rate of fat oxidation in *Agl^−/−^* as compared to WT mice (red bars). On the other hand, carbohydrate burning was compromised in *Agl*^−/−^ mice as compared to WT with (red bars) or without (black bars) GHF-201 treatment (Figure 4(A6)). However, GHF-201 increased carbohydrate burning in *Agl^−/−^* mice, where it was compromised, strongly suggesting that the compound can increase glycogen catabolism. This is a therapeutic advantage as the accumulated, poorly digested, and malconstructed glycogen in GSDIII is considered a pathogenic factor. Ambulatory activity (Figure 4(B1)) was compromised in *Agl^−/−^* mice, as compared to WT, in all periods (black bars). This was corrected in *Agl^−/−^* mice by GHF-201 again only in the light (blue bars). Similar to the weight-corrected TEE (Figure 4(A2),(A4)), the ambulatory activity of *Agl^−/−^* mice was also rescued to WT level by GHF-201, but again only in the light period (red bar, no significant difference from WT after treatment). While no statistically significant difference between WT and *Agl^−/−^* mice was observed for the distance of wheel running (Figure 4(B2), black bars), GHF-201 did increase wheel running distance in *Agl^−/−^* mice in the light and overall periods (blue bars). This increase caused wheel running distance in *Agl^−/−^* mice to be significantly increased, in the light and overall periods, as compared to WT mice (red bars). Stimulation of wheel running and correction of ambulatory activity in the light (Figure 4B) are in line with open field results (Figure 2A), also acquired in the light, and with the GHF-201-mediated stimulation of carbohydrate catabolism. Together, these results suggest that GHF-201 can improve metabolic efficacy, compromised by the diseased state, at the whole animal level. Compared to WT mice, food intake (Figure 4(C1)) was not significantly modified in *Agl^−/−^* mice in the light and was reduced in the dark and overall (black bars). GHF-201 did not exert a significant effect on food intake, possibly excluding an increase in overall food intake which was masked by a large outlier effect of an animal in which GHF-201 reduced, rather than increased, food intake (Figure 4(C1), blue bars). Excluding this outlier, an increase in overall food uptake would have been similar to that observed in the GSDIV mouse model [20]. GHF-201 also did not reverse the reduction in food intake in *Agl^−/−^* mice compared to WT mice (red bars). Water uptake, on the other hand, also reduced in all periods in *Agl^−/−^* mice compared to WT (Figure 4(C2), black bars), was corrected by GHF-201 in *Agl^−/−^* mice in the dark and overall (blue bars). Nevertheless, this correction could not overcome the reduced extent of water intake in *Agl^−/−^* mice as compared to WT (red bars). As opposed to food intake, the effect of GHF-201 on water intake was similar in direction and in statistical significance to its effect in GSDIV mice [20]. The GHF-201-mediated increase in water intake was mainly observed in the dark (and, subsequently, overall), where activity and water (and food) consumption are normally increased in mice. A GHF-201-mediated increase in water intake with a parallel increase in fuel burning (Figure 2A) constitutes an improvement in metabolic efficiency. Please note that both food and water intake plots are cumulative and thus diverge from, rather than converge with, one another with time.

### 3.5. Muscle Glycogen Quantification

Blinded quantification by transmission EM (TEM) of gastrocnemius muscle sections showed that in muscle sections derived from GSDIII mice, a higher percentage of the area was occupied by dense glycogen granules as compared to muscle sections derived from wild type mice (Figure 5A). Myofibrils in GSDIII muscle sections were also more dissociated than wild type myofibrils due to the increased glycogen storage (Figure 5B). Both increase in glycogen granule area and myofibril dissociation were significantly corrected (even though not to wild type levels) by the GHF-201 treatment. Additional TEM images are shown in Appendix A. These results were corroborated by a biochemical analysis of glycogen content in the same muscle sections (Figure 5C).

### 3.6. Glycogen Levels, Lysosomal and Mitochondrial Features of GSDIII Patient-Derived Skin Fibroblasts

Consistent with the results in gastrocnemius muscle, in skin fibroblasts derived from GSDIII patients, GHF-201 also reduced glycogen levels and rendered them similar (not statistically different) to those of HC fibroblasts, as determined by periodic acid-Schiff’s reagent (PAS) staining (Figure 6A). This result suggests that the mechanism of action by which GHF-201 alleviated GSDIII symptoms in mice is autolysosomal degradation of glycogen, which is known to be mediated by GHF-201 in GSDIV [20] and GSDIa [21].

To confirm GHF-201 mechanism of action at a cellular level, we decided to test its effects on lysosomal and mitochondrial features in fibroblasts derived from GSDIII patients. Since GHF-201 is an autophagic and catabolic activator, we anticipated that its significant effect at a cellular level will be at the lysosomal–mitochondrial axis, as shown, for instance, in GSDII [33]. Our results (Figure 6B) show that under starvation, which induces autophagic flux, GHF-201 reduced the lysosomal area, bringing it close to the area observed in HC fibroblasts. Reduced lysosomal area was also observed in other systems in healthy as compared to lysosomal impaired cells [34] and is presumably associated with increased autophagic flux and lysosomal function, which can clear overladen lysosomes from their excess content and thus reduce their size, which is increased by the lysosomal swelling this excess content causes. Mitochondrial depolarization was observed in GSDIII patient fibroblasts as compared to HC fibroblasts. This mitochondrial depolarization is probably associated with the limited glycogen degradation and carbohydrate fuel production characteristic of GSDIII (Figure 4A). In accordance with an increase in mitochondrial fueling by enhanced sutophagic catabolism, GHF-201 treatment repolarized mitochondrial membrane potential, bringing it closer to that of HC fibroblasts (Figure 6C).

## 4. Discussion

The long-term aim of this study was to test the capacity of GHF-201 to alleviate disease symptoms in the *Agl^−/−^* mouse model of GSDIII. GHF-201 specifically binds the lysosomal membrane protein LAMP1 [20]. GHF-201 was further able to re-acidify the aberrant alkaline lysosomal pH. In adult GSDIV (APBD) models, GHF-201 ameliorated the diseased state by enhancing lysosomal glycogen catabolism and autophagy.

GSDIII starts with liver involvement followed by storage of glycogen in skeletal muscle, provoking a myopathy. The disease is caused by cytoplasmic glycogen debranching enzyme (GDE) deficiency leading to accumulation of glycogen due to its reduced degradation. Glycogen accumulation in muscle is more extensive in GSDIII as compared to the other myopathy-implicating GSDs GSDII, GSDV, GSDIX, GSDX, and GSDXV [11]. In particular, human muscle biopsies from GSDIII patients show a typical and constant vacuolar myopathy, characterized by multiple and variably sized vacuoles filled with PAS-positive material [9]. Phosphorylase-limit dextrin (PLD), the type of glycogen which accumulates in GSDIII, is structurally abnormal, containing shorter outer branches [12]. While PLD is sensitive to diastase digestion and does not form polyglucosan bodies [12], it is still associated with large vacuoles and can cause histological damage to myofibrils [13,18]. In fact, as also shown by our results (Figure 5), disruption of myofibrillar structure by glycogen vacuoles is more pronounced in GSDIII than in other GSDs with prominent muscular involvement [11].

After the liver disorder, myopathy pursues [15], with occasional cardiomyopathy [16], and at later stages liver cirrhosis, and hepatocellular adenoma [17]. Interestingly, possibly due to higher overall glycogen accumulation, GSDIII manifests with excess glycogen in both cytoplasm and lysosomes, as opposed to GSDII, where excess glycogen is primarily found in lysosomes due to the aberrant function of GAA [35]. The main pathogenic factor in GSDIII, as revealed by ultrastructural analysis of muscle specimens from patients, is autophagosome accumulation. Concordantly, Laforet et al., using electron microscopy, showed the presence of large non-membrane bound sarcoplasmic deposits of normally structured glycogen as well as smaller rounded sac structures lined by a continuous double membrane containing only glycogen, corresponding to autophagosomes. A consistent SQSTM1/p62 decrease and beclin-1 increase in human muscle biopsies suggested autophagic dysregulation [9]. The latter can impede the degradation of dysfunctional organelles and proteins, and, notably, also of glycogen, in a specific pathway mediated by the starch binding domain-containing protein 1 (Stbd1), which tags glycogen for its autophagy, interacting with autophagy machinery and glycogen-related proteins, including GDE, in other domains [36]. Interestingly, among these glycogen-related proteins, GDE has the highest number of Atg8 binding motifs [36], also suggesting an important role of autophagy in GSDIII, which might take place if GDE deficiency impairs the interaction with the autophagy machinery. Moreover, dysfunctional autophagy plays a pivotal role in disrupting muscle homeostasis and causing structural damage to sarcomeres [37]. The tight link between autophagic impairment and GSDIII pathology can, thus, explain the beneficial effects of GHF-201, a general autophagic activator [20], in improving motor function (Figure 2) and skeletal muscle health (Figure 5) in a GSDIII mouse model. Other possible systemic effects of GHF-201, unrelated to autophagy enhancement, probably do not contribute to the beneficial effect of GHF-201, as it did not influence animal weight gain and histology in the long-term and behavior (Irwin test) in the short-term [20]. Even though the effect of GHF-201 on motor and behavioral features was not tested in WT mice, we assume that the beneficial effect of GHF-201 is specific to *Agl^−/−^* mice because PLD accumulation in this strain disrupts muscle structure (Figure 5 and Appendix A). Therefore, partial removal of this pathogenic PLD in *Agl^−/−^* mice can more noticeably relieve the motor dysfunction caused by the disrupted muscle structure than in WT mice where such a disruption and consequent motor dysfunction do not exist. However, since autophagy and autophagic catabolism are generally considered cell protective and energy producing, GHF-201 might possibly improve animal health in general, even in WT mice.

Our blood biochemistry panel demonstrated that GHF-201 treatment led to improved liver and muscle function and partially corrected blood hypoglycemia and hypertriglyceridemia (Figure 3), also observed in GSDIII patients [38]. Both hypoglycemia and hyperlipidemia are expected outcomes of glycogen surcharge since more glucose is shunted from blood to intracellular glycogen and since high glycogen levels can inhibit the energy sensor AMPK [39,40,41,42], leading to activation of acetyl CoA carboxylase, the rate-limiting enzyme of fatty acid synthesis, which is phospho-inhibited by AMPK. Reduction of glycogen levels by GHF-201 (Figure 5) expectedly mitigated this hyperlipidemia. The blood lipid profile can also reflect fuel utilization. As glycogen mobilization is compromised in GSDIII and enhanced by GHF-201, we expected that GSDIII mice use lipid instead of carbohydrate fuel as a compensatory reaction. However, as suggested by the hypertriglyceridemia observed in untreated GSDIII mice (Figure 3), the mice are not capable of this metabolic compensation. Notably, as revealed by both indirect calorimetry (Figure 4) and the blood biochemistry panel (Figure 3), GHF-201 treatment enabled this metabolic compensation by boosting fat burn in the light (Figure 4), which is possibly reflected in the reduction in blood triglyceride levels (Figure 3). A similar effect of GHF-201 was observed in the GSDIV modeling *Gbe^ys/ys^* mice [20]. Interestingly, GHF-201-mediated improvement of fat oxidation (Figure 4A), and other in vivo metabolic effects (TEE, (Figure 4A), ambulatory activity (Figure 4B), and wheel running (Figure 2B)) were more pronounced in the light than in the dark. Since mice are nocturnal animals, in which food is mostly consumed in the dark (see also Figure 4C) and digested in the light, it can be conjectured that GHF-201, affecting fuel digestion, or autophagic substrate catabolism, exerts a larger effect in the light. Along these lines, a possible explanation for the GHF201-associated decrease of activity in the dark is that the untimely stimulation of fuel-generating catabolism in the dark (Figure 4(B1)) might lead to fuel excess known to be detrimental as it might lead to mitochondrial fragmentation in pancreatic beta cells [42] and unbalanced reactive oxygen species (ROS) production [43]. However, the fact that GHF-201 increased ambulatory activity in the dark in GSDIV modeling mice [20], while decreasing it in GSDIII modeling mice (Figure 4(B1)), calls for an explanation. One important difference between these two diseases is that in GSDIV patients, and in GSDIV-modeling mice, polyglucosan bodies (PB) formed by brancher deficiency are architecturally different from phosphorylase-limit dextrin in GSDIII formed by debrancher deficiency. The large size and unique amylopectin architecture of PB in GSDIV models causes them to clog axons, which is not the case for the aberrant glycogen (phosphorylase-limit dextrin) in GSDIII. Therefore, PB-clearance from axons and peripheral nerve by GHF-201 in a GSDIV mouse model [20] might enhance mobility by improving nerve–muscle communication. This clearance of the noxious PB from axons and neuromuscular junctions might override the putative detrimental effect of excess fuel in the dark and so might be more important for locomotion and enhance it also in the dark in GSDIV mice, as opposed to GSDIII mice.

In conclusion, we show here that GHF-201, a LAMP-1-targeting small molecule acting as a general activator of autophagic flux, can partially correct aberrant metabolic, motor, and myopathological phenotypes in the *Agl^−/−^* mouse model of GDE deficiency, or GSDIII, a disease implicating a particularly high glycogen burden, causing liver and myofibrillar injury.

## Figures and Tables

**Figure 1 biomolecules-14-00893-f001:**
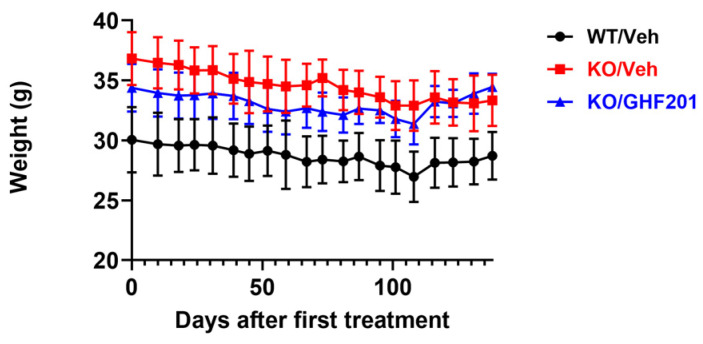
Weight over time in WT mice, *Agl^−/−^* (KO) mice injected intramuscularly biweekly with 10% DMSO solvent control, and KO mice injected in the same way with 250 mg/kg GHF-201. n = 8 WT/Veh, n = 6 KO/Veh, and n = 10 KO/GHF-201 mice were used. Throughout the period, KO mice were heavier than WT mice (*p* < 0.0001, Two Way ANOVA with repeated measures). Due to a >10% weight reduction between biweekly weightings, one mouse was removed from the KO/Veh group on day 116 after treatment initiation and one mouse was removed from the KO/GHF201 group on day 52 after treatment initiation.

**Figure 2 biomolecules-14-00893-f002:**
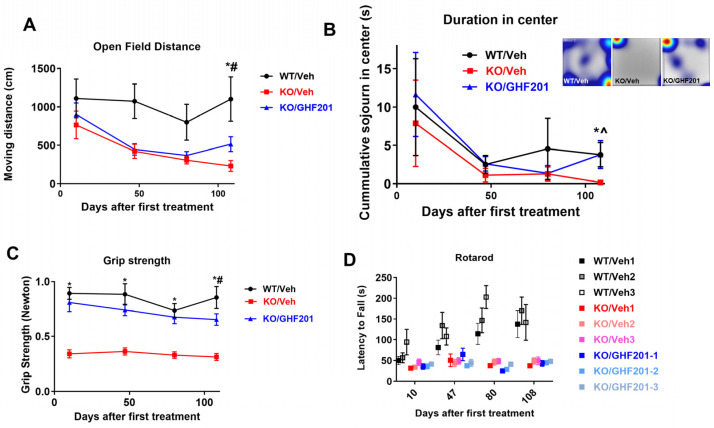
Movement distance (**A**), sojourn in the center of an open field arena (inset representative open field sojourn heatmaps) (**B**), grip strength (**C**), and latency to fall from a rotating rod (**D**) were measured in n = 8 *Agl^−/−^* mice treated with 10% DMSO vehicle (KO/Veh), n = 6 wild type mice treated with vehicle (WT/Veh), and n = 10 *Agl^−/−^* mice treated with GHF-201 (KO/GHF-201). In (**D**), each rotarod session included three consecutive runs (1–3), separated by 10 min pauses. Two Way ANOVA with repeated measures shows that only in (**C**) GHF-201 treatment values were higher than vehicle-treated values in KO mice throughout the period (*p* < 0.05). Linear regression analysis of overall latency over time (**D**), shows that WT animals developed a significantly higher learning capacity than KO/Veh and KO/GHF-201 mice. Training capacity, shown by increase in latency over runs, was only demonstrated in WT mice 80 days post-treatment initiation. See text for statistical analysis. *, KO/GH201 is significantly different from KO/Veh; *#, KO/GHF-201 is significantly different from both KO/Veh and WT/Veh; *^, KO/GHF-201 is significantly different only from KO/Veh and not from WT/Veh (correction effect). Difference significance determined by multiple *t*-tests with Sidak post hoc correction. All error bars represent s.e.m.

**Figure 3 biomolecules-14-00893-f003:**
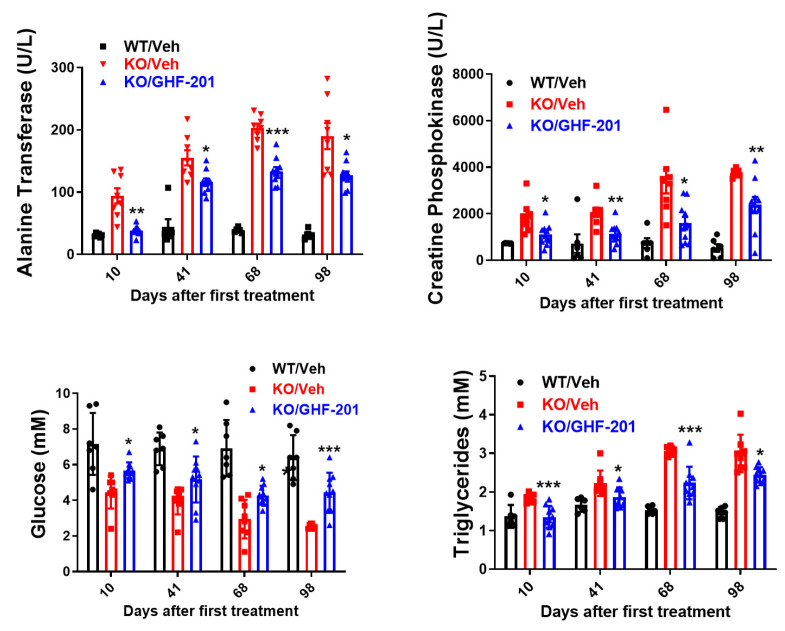
Blood metabolic panel based on n = 8 *Agl^−/−^* mice treated with 10% DMSO vehicle (KO/Veh), n = 6 wild type mice treated with vehicle (WT/Veh), and n = 10 *Agl^−/−^* mice treated with GHF-201, as indicated. At all time points, GHF-201 significantly reduced alanine transferase and creatine phosphokinase, demonstrating partial restoration of liver and muscle damages, respectively. Additionally, blood glucose was increased, and blood triglycerides were decreased by GHF-201 in treated *Agl^−/−^* mice. Statistical significance between KO/Veh and KO/GHF-201 at the different time points was determined by Two Way ANOVA with repeated measures with a Tukey post hoc test according to simple effects within rows (i.e., times). *, 0.01 < *p* < 0.05; **, 0.001 < *p* < 0.01; ***, *p* < 0.001.

**Figure 4 biomolecules-14-00893-f004:**
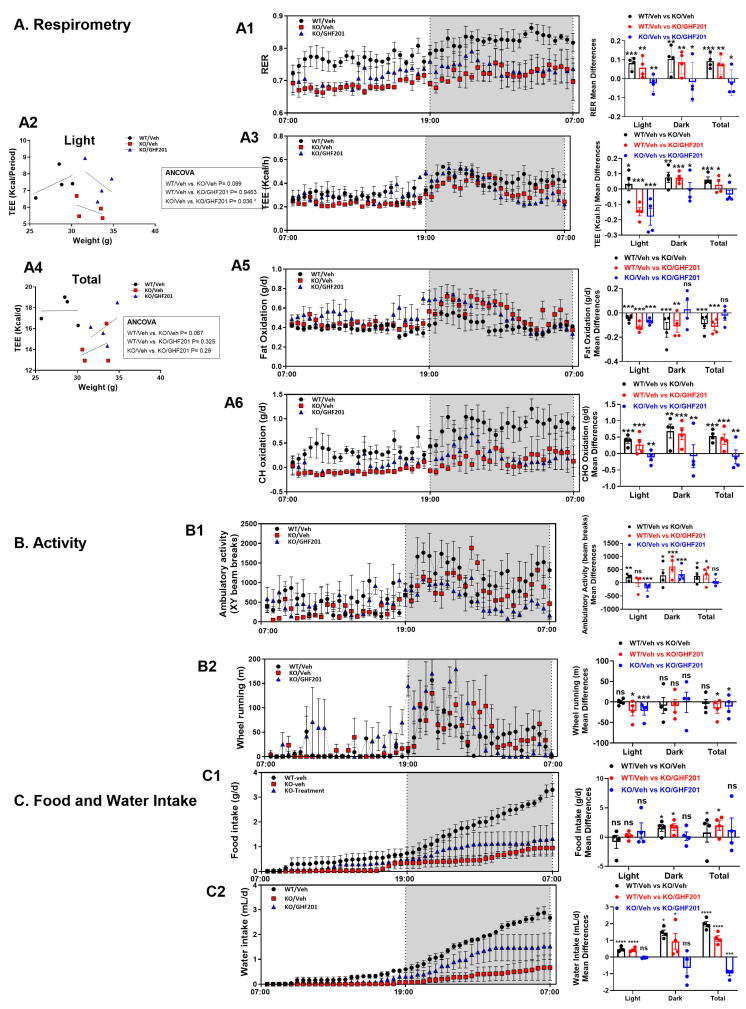
Indirect calorimetry and metabolic cage analysis. Following a 48 h habituation period, mice (n = 4 from each study arm indicated) were monitored over a 24 h period. Data are mean ± SEM from eight-month-old mice. ((**A**)—Respirometry) As demonstrated by the positive mean differences, in the light, dark, and overall (total), vehicle-treated *Agl^−/−^* mice demonstrate lower respiratory exchange ratio (RER, (**A1**)), total energy expenditure (TEE, (**A2**–**A4**)), and carbohydrate oxidation (**A6**). As shown by the negative mean differences, fat oxidation was reduced in *Agl^−/−^* mice in light, dark, and overall as compared to WT mice (**A5**). All these parameters were increased or corrected by GHF-201 at all times, except for fat oxidation in the dark and overall (**A4**). ANCOVA results suggest that weight-dependent TEE is reduced in *Agl^−/−^* compared to WT mice in both the light condition (**A2**) and overall (**A4**), and that GHF-201 corrects this TEE reduction in *Agl*^−/−^ mice in the light. ((**B**)—Activity) Ambulatory activity (**B1**) and wheel running (**B2**) were also increased (corrected), or not affected, by GHF-201 as indicated. ((**C**)—Food and Water Intake) Food intake (**C1**) was not significantly affected, and water intake ((**C2**), total cumulative values are shown in the bar graphs) was increased, mainly in the dark period, by GHF-201. Statistical significance of mean differences was determined by One Way ANOVA with repeated measures with a Tukey post hoc test. *, 0.01 *< p* < 0.05; **, 0.001 < *p* < 0.01; ***, *p* < 0.001; ****, *p* < 0.0001; ns, non-significant.

**Figure 5 biomolecules-14-00893-f005:**
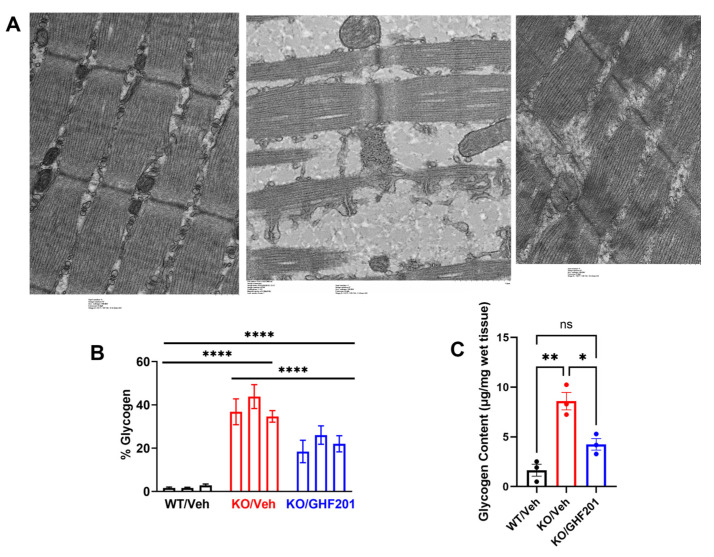
(**A**) Representative transmission electron microscopy images of longitudinal sections of *gastrocnemius* muscle collected from 8-month-old animals treated as indicated. Note granular glycogen material interspersing sarcomeres in the section from untreated *Agl^−/−^* mouse. Also note variable width of sarcomeres in *Agl^−/−^* mouse sample. These ultrastructural phenotypes were partially corrected in GHF-201-treated *Agl^−/−^* mouse. (**B**) Quantification of area percent of glycogen based on analysis of TEM images from n = 3 animals. ****, significant differences between arms (*p* < 0.0001, Two Way ANOVA (factors are “animal” and “arm”) with Tuckey post hoc correction). (**C**) Biochemical quantification of *Gastrocnemius* glycogen. Glycogen levels in muscle tissues collected from n = 3 mice/arm were determined by Merck’s glycogen assay kit and statistically analyzed by One Way ANOVA with Tuckey post hoc correction. *, *p* < 0.05; **, *p* < 0.01, ns, *p* < 0.08.

**Figure 6 biomolecules-14-00893-f006:**
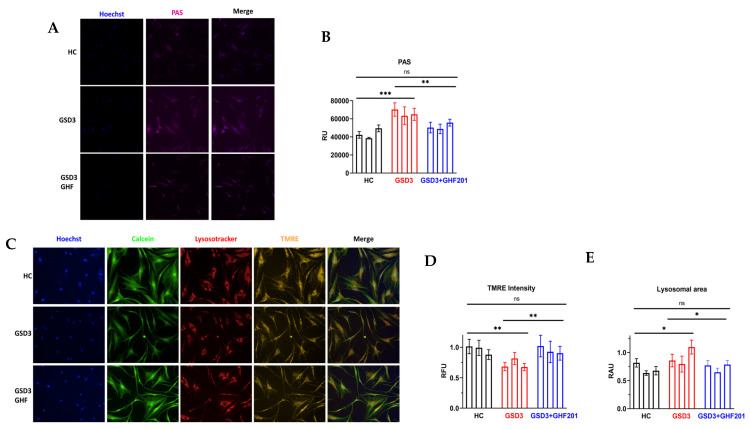
Skin fibroblasts from 3 GSD3 and 3 HC individuals were cultured in 96-well plates for 48 h in starvation medium without FBS and glucose. Subsequently, cells were live-stained with (**A**,**B**) PAS reagent to quantify intracellular glycogen, or (**C**–**E**) a mix of fluorescent dyes which included Hoechst, Calcein-AM, TMRE, and Lysortracker red to, respectively, stain nuclei (blue), cytoplasm (green), respiring mitochondria (yellow), and lysosomes (red). Multiple images from the live-stained cells were automatically obtained by an Operetta G1 image analyzer under environmentally controlled conditions. Shown are representative images and their quantification in relative units. Statistical significance of differences was determined by Two Way ANOVA with Tuckey post hoc correction. *, *p* < 0.05; **, 0.001 < *p*< 0.01; ***, *p* < 0.001; ns, not significant.

## Data Availability

Data is contained within the article. The original contributions presented in the study are included in the article. Further inquiries can be directed to the corresponding authors.

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
