# Peer review of "The Autophagic Activator GHF-201 Can Alleviate Pathology in a Mouse Model and in Patient Fibroblasts of Type III Glycogenosis"

_biomolecules, 2024, doi:10.3390/biom14080893_

Round 1

Reviewer 1 Report

Comments and Suggestions for Authors

The authors investigate the ability of an autophagy enhancing drug, GHF201, to ameliorate the effects of glycogen debranching enzyme deficiency in mouse models and patient fibroblasts. GHF-201 was previously shown to improve a GSD4 mouse model (Kakhlon et al. 2021). Here, seems to partially improve some behavioral and biochemical parameters affected in Agl-/- mice. The present study as well as the previous study would benefit from showing how the drug impacts all measured parameters in WT. This would allow to distinguish between more general systemic effects of the drug (especially behavioral effects) and those specifically resulting from increased autophagic flux. It is apparent that treated Agl-/- mice consume more food, gain more weight, and have more grip strength. Several other parameters that were affected by the drug could be simply consequences of increased food intake. Several parameters are only very slightly changed after treatment. For instance, the in vivo metabolic profiles show very small differences, and insufficient description of methodological detail does not exclude that statistical significance maybe reached due to the use of inappropriate ways of analyses. Similar issues exist with the analyses and interpretation of the fibroblast high content data. The quantification of muscle glycogen by TEM seems unusual. Especially, the reduction of glycogen content needs corroboration as it is the only indication that GHF201 acts truly to mitigate the core problem that exists in GSDIII, namely the inability to degrade phosphorylase-limit glycogen. Overall, the paper does not fully convince this reviewer that GHF201 helps degrade phosphorylase-limit glycogen to an extent that it would be therapeutically relevant.  

Major concerns:

Line 134: please explain in detail by which statistical methods outliers were identified.

Line 101: please explain whether WT controls and Agl-/- mice were littermates, i.e. derived from het/het breedings or from independent breedings of Agl-/- with Agl-/- for the generation of KOs and WT with WT for the generation of WT controls.

Line 154: Please state whether the observed weight difference is statistically significant. Is it possible that weight differences are caused by non-littermate comparisons? See comment for line 101. If so, please include this as a possible explanation in section 3.1.

Figure 1: In this figure weight has been determined every 6-7 days. This is inconsistent with the description in the method part, which indicates biweekly weight monitoring. Please consolidate. Please include information on the number of mice used in each average, and whether (and when and from which group) any mice were removed due to >20% weight loss. In addition the weight at treatment initiation is not recorded, precluding the authors to evaluate weight in comparison to treatment initiation (this is one criterion for trial exclusion mentioned in the method section line 106). The authors should clarify this in the text.

Figure 2: Please include data at study initiation.

Figure 2D: This figure is hard to interpret, especially due to the stacking of the data series. Also, there are no statistical analyses indicated that support the authors’ claim that Wild type mice manifest training and motor learning capacity, but KO mice do not. Please use a different representation of the data, where (1) latencies are not stacked, (2) consecutive trials are next to each other and grouped by day post treatment, (3) error bars remain included, and (4) statistical test results are added. The drawing below may provide a helpful starting point (drawing only included in the submitted review PDF).

Line 173-176: The authors mention a weight drop of all mice by >10% from the previous weight reading. This is inconsistent with weight recordings in Fig.1, where no average weight drop of >10% around post treatment day 100 can be observed.

Fig. 3: please include individual data points for each bar. Given that more than 2 groups are compared, simple T-tests are not appropriate. Please conduct ANOVA with appropriate post-hoc tests and multiple comparison adjustment of p-values. Indicate which test is used and state the significance threshold.

Fig.4: The organization of the figure is too rough. Please substructure the figure by assigning more panel letters. Calculation of the averages in the bar charts is not explained. Please show individual data points. Those should be 4 in each bar, as n=4. The time points should first be averaged per mouse. Then averages should be calculated across the four biological replicates. Statistical tests used in the bar charts are not explained. They should be ANOVA with appropriate post-hoc. Significance threshold must be declared.

Re In vivo metabolic profile: Please include detailed descriptions of the XYZ beam and wheel running assays used. The information in the referenced publication by Kakhlon et al. 2021 (EMBO Mol Med) is not sufficient to understand the method. The authors should discuss how GHF201 can lead to increased ambulatory activity in the light and decreased activity in the dark. In addition the authors should explain why the same drug leads to increased ambulatory activity in the dark in GBE1-deficient mice (Kakhlon et al. 2021).

Line 215-221: It is questionable whether EM is the best method to quantify muscle glycogen accumulation and rescue thereof in mouse muscle. EM provides only very small fields of view and, to avoid conclusions drawn from non-representative sections and images, several sections and images per mouse should be analyzed. Please explain (in the method part) how many sections from how many animals were analyzed. Quantified areas from different sections of the same animal should be averaged first. Each animal should be represented by only one data point in Fig. 5B. The EM image shown for the non-treated GSD3 mouse looks very different in format section plane and overall contrast from the other two, which makes this reviewer question whether the authors selected comparable sections. The authors should include additional images from different mice in the supplement. Furthermore, the authors should corroborate their EM data by biochemical quantification of glycogen.

Fig. 6: Please add panel letters. As analyses were conducted with n=3, averages should be calculated across the three biological replicates, not across all individual images taken. This dictates that bars should only contain 3 data points each, representing the level of biological replication. Statistical significance should be calculated on the basis of n=3 after averaging of images taken from the same biological replicate. The method section related to this figure mentions that outliers were removed. The authors should describe the method they used to identify outliers. Please also describe the statistical tests used.

Minor concerns:

Line 53: It is not common to speak of GP as a phosphorylating enzyme. Maybe better “GP phosphorylytically releases outer glucose residues in the form of glucose 1-phosphate.”

Line 60: add a citation for glycogen degradation.

Line 67: add a citation for abnormal glycogen metabolism in diabetes.

Line 165-166: place commas around ‘respectively’.

Line 175: remove comma after ‘GHF-201’

Figure legend Fig. 2: The legend indicates an interval of 10min between consecutive rotarod tests, while the method section indicates 20min. Please consolidate.

Line 113: please include information on how the blood was obtained from the mice.

Line 211: enter space between ‘increased’ and ‘in’

Author Response

Reviewer 1

Comment: The present study as well as the previous study would benefit from showing how the drug impacts all measured parameters in WT. This would allow to distinguish between more general systemic effects of the drug (especially behavioral effects) and those specifically resulting from increased autophagic flux.

Reply: Thank you for this comment. The measured parameters related to accumulation of the pathogenic phosphorylase-limit dextrin  (i.e., ultrastructural changes of sarcomeres (Figure 5), and glycogen accumulation (Figure 6A)) will probably not show any response to GHF-201 in WT animals since the latter do not manifest any accumulation of phosphorylase-limit dextrin, or the glycogen vacuoles and glycogen surcharge associated with it. As for other parameters, the effect of GHF-201 on motor/behavioral parameters (importantly gait, but also positional passivity, touch escape, body position, and transfer behavior) and reflexes (extension reflex, righting reflex, trunk curl, tail elevation, pelvic elevation) in WT congenic mice at different concentrations and exposure times has already been tested and published as part of the Irwin tests (Kakhlon et al (2021) EMBO Mol Med 13:e14554). We have also showed in that paper that GHF-201, referred to as 144DG11 in that publication because it was prior to its sublicensing to the GHF company, does not significantly modify weight, weight gain over time and histopathology in these WT C57BL/6J mice, also congenic to the Agl-/- mice (Figure EV2 there). In addition, necroscopic macroscopy performed by an external contractor (Envigo) as part of GHF-201 development for clinical use, showed it has no adverse effect at an organ level. These data cannot be disclosed due to contractual discretion with GHF. Taken together, we believe that these comprehensive examinations show that GHF-201 does not have general systemic effects.

Comment: It is apparent that treated Agl-/- mice consume more food, gain more weight, and have more grip strength. Several other parameters that were affected by the drug could be simply consequences of increased food intake. Several parameters are only very slightly changed after treatment. For instance, the in vivo metabolic profiles show very small differences, and insufficient description of methodological detail does not exclude that statistical significance maybe reached due to the use of inappropriate ways of analyses. Similar issues exist with the analyses and interpretation of the fibroblast high content data. The quantification of muscle glycogen by TEM seems unusual. Especially, the reduction of glycogen content needs corroboration as it is the only indication that GHF201 acts truly to mitigate the core problem that exists in GSDIII, namely the inability to degrade phosphorylase-limit glycogen. Overall, the paper does not fully convince this reviewer that GHF201 helps degrade phosphorylase-limit glycogen to an extent that it would be therapeutically relevant.  

Reply: 

GHF-201 treatment did not cause Agl-/- mice to gain more weight (see for instance Figure 1). While GHF-201 caused a non-statistically significant trend of increased food consumption in the dark in Agl-/- mice (Figure 4C), we would combine this increased food intake with the increased fuel burning and the food requiring energy expenditure associated with the GHF-201 treatment. Put together, we would interpret these observations as an increase in food metabolization efficacy, caused by GHF-201-mediated increased autophagic catabolism. Therefore, we would maintain that food intake does not stand on its own, but should rather be interpreted in conjunction with other metabolic parameters. Such a consolidated interpretation would connect the increased food intake with the increased fuel burning required for the increased energy expenditure.

The other comments of the reviewer will be replied below as part of our replies to the Major concerns. 

Comment: Line 134: please explain in detail by which statistical methods outliers were identified.

Reply: Outliers were removed using the Prism (version 10.2.3) ROUT method with Q=10%. This information was added to the text.

Comment: Line 101: please explain whether WT controls and Agl-/- mice were littermates, i.e. derived from het/het breedings or from independent breedings of Agl-/- with Agl-/- for the generation of KOs and WT with WT for the generation of WT controls.

Reply: WT controls were Agl+/+ littermates derived from heterozygous/heterozygous breeding which also produced the Agl-/- KO mice used in this study and Agl+/- heterozygotes. We added this information to the main text.

Comment: Line 154: Please state whether the observed weight difference is statistically significant. Is it possible that weight differences are caused by non-littermate comparisons? See comment for line 101. If so, please include this as a possible explanation in section 3.1.

Reply: Yes the difference in weight over time between WT and KO mice is statistically significant. This information was added to the main text. As stated, WT mice were Agl+/+ littermates derived from heterozygous/heterozygous breeding which also produced the Agl-/- KO mice used in this study.

Comment: Figure 1: In this figure weight has been determined every 6-7 days. This is inconsistent with the description in the method part, which indicates biweekly weight monitoring. Please consolidate. Please include information on the number of mice used in each average, and whether (and when and from which group) any mice were removed due to >20% weight loss. In addition the weight at treatment initiation is not recorded, precluding the authors to evaluate weight in comparison to treatment initiation (this is one criterion for trial exclusion mentioned in the method section line 106). The authors should clarify this in the text.

Reply: According to the IACUC approval of this protocol, animal weights have to be recorded biweekly to monitor weight deviations which can indicate health deterioration. This is the reason for indicating biweekly weight recording in the Methods section. In practice, we increased the weighting frequency to once a week so that it can be reported as a result and so that we will have enough data points for statistical analysis. We corrected the weighting frequency in the Methods section.

Mouse numbers were added to the legend of Figure 1.

We added the number, times and groups of mouse removal due to excessive weight loss (it was actually >10% in two consecutive biweekly measurements, rather than >20% from treatment initiation).

We have measured the locomotor parameters only at the indicated times after treatment initiation. Our goal was to compare Agl-/- treated with Agl-/- untreated mice using WT mice as a reference. To that end we believe that comparisons made at fixed times after treatment initiation are sufficient, as we are not comparing change from baseline, but rather the change, or KO response to treatment, itself. The fact that this change is compared at the same time points after treatment initiation can exclude time as a factor influencing the response to the treatment. This experimental setup, where movement in an open field is measured at fixed time points after treatment initiation (and not at initiation itself) is also used in several high impact publications, e.g., Zhai et al (2023) Gene Therapy 30: 75; van Buel et al (2015) J Neuroinfl 15: 232; Li et al (2018) Biol Reproduction 99: 349; and also our own publication on GSD type IV Kakhlon et al (2021) EMBO Mol Med 13:e14554. 

Comment: Figure 2: Please include data at study initiation.

Reply: Weight at initiation of treatment was added.

Comment: Figure 2D: This figure is hard to interpret, especially due to the stacking of the data series. Also, there are no statistical analyses indicated that support the authors’ claim that Wild type mice manifest training and motor learning capacity, but KO mice do not. Please use a different representation of the data, where (1) latencies are not stacked, (2) consecutive trials are next to each other and grouped by day post treatment, (3) error bars remain included, and (4) statistical test results are added. The drawing below may provide a helpful starting point (drawing only included in the submitted review PDF).

Reply: We have corrected the figure as per the reviewer suggestion and added statistical analyses which indeed added more details about the differences between the different groups. We thank the reviewer for these suggestions.

Comment: Line 173-176: The authors mention a weight drop of all mice by >10% from the previous weight reading. This is inconsistent with weight recordings in Fig.1, where no average weight drop of >10% around post treatment day 100 can be observed.

Reply: We guess the reviewer refers to lines 181-183 because weight drop of mice is not mentioned in lines 173-176. We mention that the weight of all mice was decreased by >10% after the 4th test. This decrease is indeed not shown in Figure 1 because by the time it took place at day 154 after treatment initiation (N.B., not around day 100, the previous plot continued until day 138), we have already discontinued the experiment and were not able to perform the 5th test. Originally, the weight over time plot was meant to encompass tests times. We have now added these additional two time points even though the 5th test did not take place.

Comment: Fig. 3: please include individual data points for each bar. Given that more than 2 groups are compared, simple T-tests are not appropriate. Please conduct ANOVA with appropriate post-hoc tests and multiple comparison adjustment of p-values. Indicate which test is used and state the significance threshold.

Reply: We included data from individual animals in each bar. Statistical significance analyses (comparisons between GHF-201-treated and untreated Agl-/- mice) were now conducted by Two Way ANOVA with repeated measures with Tukey post-hoc test using simple effects within rows (times) analysis. This is indicated in the revised legend of Figure 3. 

Comment: Fig.4: The organization of the figure is too rough. Please substructure the figure by assigning more panel letters. Calculation of the averages in the bar charts is not explained. Please show individual data points. Those should be 4 in each bar, as n=4. The time points should first be averaged per mouse. Then averages should be calculated across the four biological replicates. Statistical tests used in the bar charts are not explained. They should be ANOVA with appropriate post-hoc. Significance threshold must be declared.

Reply: The figure was subdivided to more panels as requested. Averaging the time points per mouse and comparing them by normal ANOVA could be problematic because time itself also affects the results or contributes to variability (p<0.1, Two Way ANOVA analysis performed to demonstrate this point (i.e., not a part of the manuscript)). Therefore, we performed instead of ANOVA, ANOVA with repeated measures to account for the effect of time and compare the different courses of change in the means, rather than the means themselves. This approach for the analyzing indirect calorimetry in metabolic cages was also taken by other workers, such as John et al. (2022) Commun Biol 5: 946 and Adriaenssens et al. (2023) JCI Insight 8: e164921. We also show the results as differences in means, which are the output of repeated measures ANOVA, between WT/Veh vs KO/Veh to show the disease effect, WT/Veh vs KO/GHF-201 to show whether GHF201 treatment brings the parameter tested closer to WT levels, suggesting amelioration, and KO/Veh vs KO/GHF-201 to test treatment effect. According to the mouse numbers, n=4 mean differences are shown for each bar in the bar graphs summarizing the longitudinal data. This new statistical analysis changed some of the results compared to the previous statistical analysis, which was based on pooling all data points in a group. However, the improvement of all metabolic parameters by GHF-201 according to the previous analysis is sustained in this new analysis. Analysis was performed by One Way ANOVA (to analyze the entire time-dependent trend, rather than each separate time point in Two Way ANOVA) with repeated measures and Tukey post-hoc test and p values are now indicated.

Comment: Re In vivo metabolic profile: Please include detailed descriptions of the XYZ beam and wheel running assays used. The information in the referenced publication by Kakhlon et al. 2021 (EMBO Mol Med) is not sufficient to understand the method. The authors should discuss how GHF201 can lead to increased ambulatory activity in the light and decreased activity in the dark. In addition the authors should explain why the same drug leads to increased ambulatory activity in the dark in GBE1-deficient mice (Kakhlon et al. 2021).

Reply: Thank you for the comment. Explanations on the XYZ beam and wheel running systems was added in section 2.1.

As for the light-selective ameliorative effect of the GHF201 effect, this is explained in the one before last paragraph of the Discussion: “Interestingly, GHF-201-mediated improvement of fat oxidation (Figure 4A), and other in vivo metabolic effects (TEE, (Figure 4A), ambulatory activity (Figure 4B), wheel running (Figure 2B)) were more pronounced in the light than in the dark. Since mice are nocturnal animals, in which food is mostly consumed in the dark (see also Figure 4C) and digested in the light, it can be conjectured that GHF-201, affecting fuel digestion, or autophagic substrate catabolism, exerts a larger effect in the light.”

Following the reviewer comment, we also added the following text there: “Along these lines, a possible explanation for the GHF201-associated decrease of activity in the dark is that the untimely stimulation of fuel-generating catabolism in the dark (Figure 4B1) might lead to fuel excess known to be detrimental as it might lead to mitochondrial fragmentation in pancreatic beta cells (e.g., Alan & Scorrano (2022) Current Biology 32, R589–R683) and unbalanced reactive oxygen species production (Fisher-Wellman et al (2013) Free Radic Biol Med 65:1201).” 

Regarding the fact that GHF-201 increased ambulatory activity in the dark in GSDIV modeling mice, while decreasing it in GSDIII modeling mice, this difference can be explained in several ways. However, one important difference between these two diseases is that in GSDIV patients, and in GSDIV-modeling mice, polyglucosan bodies (PB) formed by brancher deficiency are architecturally different than phosphorylase-limit dextrin in GSDIII formed by debrancher deficiency. The large size and unique amylopectin architecture of PB in GSDIV models causes them to clog axons, which is not the case for the aberrant glycogen (phosphorylase-limit dextrin) in GSDIII. Therefore, PB-clearance from axons and peripheral nerve by GHF201 in a GSDIV mouse model (Kakhlon et al (2021) EMBO Mol Med 13:e14554) might enhance mobility by improving nerve-muscle communication. This clearance of the noxious PB from axons and neuromuscular junctions might override the putative detrimental effect of excess fuel in the dark and so might be more important for locomotion and enhance it also in the dark in GSDIV mice, as opposed to GSDIII mice.

Comment: Line 215-221: It is questionable whether EM is the best method to quantify muscle glycogen accumulation and rescue thereof in mouse muscle. EM provides only very small fields of view and, to avoid conclusions drawn from non-representative sections and images, several sections and images per mouse should be analyzed. Please explain (in the method part) how many sections from how many animals were analyzed. Quantified areas from different sections of the same animal should be averaged first. Each animal should be represented by only one data point

Reply: EM has already been used to quantify muscle glycogen in GSIII mouse models and patients (Laforet et al (2019) Acta Neuropathol Commun 7: 167; Vidal et al (2018) Mol Ther 26: 890). This method in muscle is also described in a JoVE article (https://app.jove.com/v/63347/quantification-of-subcellular-glycogen-distribution-in-skeletal-muscle-fibers-using-transmission-electron-microscopy). The quantification described here is based on 10 sections per muscle, which are considered representative of a muscle fiber which they sample across its entire area (https://app.jove.com/v/63347/quantification-of-subcellular-glycogen-distribution-in-skeletal-muscle-fibers-using-transmission-electron-microscopy).

Comment: The EM image shown for the non-treated GSD3 mouse looks very different in format section plane and overall contrast from the other two, which makes this reviewer question whether the authors selected comparable sections. The authors should include additional images from different mice in the supplement. Furthermore, the authors should corroborate their EM data by biochemical quantification of glycogen.

Reply: All TEM images were acquired at the same magnification, acceleration voltage, and emission amperage. Therefore, all images are comparable. The reason for the different appearance of GSD3 mouse muscle images is biological and reflects disruption of the muscle fibers by the glycogen granules. A similar visual effect of the GSD3 pathology in TEM images of muscle fibers was observed in GSD3 patients (Laforet et al (2019) Acta Neuropathol Commun 7: 167) and murine muscle (Vidal et al (2018) Mol Ther 26: 890). As requested, additional figures were added to a supplemental figure.

We accept the reviewer’s critique that glycogen in the muscle fibers should also be quantified biochemically. We added data of total glycogen levels in muscle samples from the same animals analyzed by TEM experiments. We also updated the Methods section accordingly. These results show the same trend as the TEM results and thus support them.

Comment: Fig. 6: Please add panel letters. As analyses were conducted with n=3, averages should be calculated across the three biological replicates, not across all individual images taken. This dictates that bars should only contain 3 data points each, representing the level of biological replication. Statistical significance should be calculated on the basis of n=3 after averaging of images taken from the same biological replicate. The method section related to this figure mentions that outliers were removed. The authors should describe the method they used to identify outliers. Please also describe the statistical tests used.

Reply: Thank you for these comments. Panel letters were added. Data from each subject (patients and healthy controls) are now presented separately and statistical significance was recalculated by Two Way ANOVA with variability among images from each subject as one of the factors and intergroup variability as the other. This information was added to the figure legend. As mentioned above, as a reply to another question from this reviewer, outliers were removed using the Prism (version 10.2.3) ROUT method with Q=10%. This information was added to the Methods section under statistical analysis.

Minor concerns:

Comment: Line 53: It is not common to speak of GP as a phosphorylating enzyme. Maybe better “GP phosphorylytically releases outer glucose residues in the form of glucose 1-phosphate.”

Reply: Corrected.

Comment: Line 60: add a citation for glycogen degradation.

Reply: Done. We have added Adeva-Andany et al (2016) BBA Clin 5: 85–100.

Comment: Line 67: add a citation for abnormal glycogen metabolism in diabetes.

Reply: Done. We have added Brereton et al (2016) Nat Commun 13496 and Ashcroft et al (2017) Cell Metabol 26: 17.

Comment: Line 165-166: place commas around ‘respectively’.

Reply: Done. Currently in line 184.

Comment: Line 175: remove comma after ‘GHF-201’

Reply: Done.

Comment: Figure legend Fig. 2: The legend indicates an interval of 10min between consecutive rotarod tests, while the method section indicates 20min. Please consolidate.

Reply: The correct time is 10 min. We corrected the Methods section. Thank you for noticing that.

Comment: Line 113: please include information on how the blood was obtained from the mice.

Reply: Added. Blood was collected by submandibular bleeding.

Comment: Line 211: enter space between ‘increased’ and ‘in’

Reply: Done.

Reviewer 2

Comment: congratulation for the great quality in the research of this disorder. The biochemical pathways and metabolic views are explained in this paper with great details. At line 98 you talk about combination therapy. I suggest to explain the current treatment in this form og glycogen storage disorder and the clinical trial with gene therapy and mRNA to face the clinical impact of this new therapy you propose.

Reply: Thank you for this positive view of our research. We added text on this clinical trial and other therapeutic options for GSDIII in combination with our autophagic stimulator GHF-201 (lines 100-110).

Comment: The comparison between studies on mice and in fibroblasts of humans are very interesting and  you explain very enthusiastic results. We understand that the GSD mouse model is very fragile and it is very complicated to manage.  Did you perform also metabolomic studies or The hipothesis of the involvement of lysosomal-mitochondrial axis also in GSDIII is very challenge to understand. The figures 5 and 6 are very clear, less the figure 2 and 3.

Reply: Since we show here that GHF-201, as an activator of autophagic flux, ameliorated GSDIII pathology, it is indeed interesting to investigate its effect on the lysosomal-mitochondrial axis. The main reason is that more efficient autophagy can also enhance mitophagy and mitochondrial quality control thus increasing the proportion of functional mitochondria. The effect of GHF-201 on this axis has been extensively studied by us in GSDIV (Kakhlon et al (2021) EMBO Mol Med 13:e14554) and GSDIa (doi.org/10.1101/2023.02.20.529109, in preparation) models. These works included bioenergetic analyses, transcriptional regulation of the NAD+/NADH-Sirt1-TFEB regulatory axis, epigenetic and proteomic studies. Therefore, as these metabolic studies should be comprehensive, for this work, a metabolic analysis of the lysosomal-mitochondrial axis extends beyond its scope and will be investigated in a separate work. We show here results in GSDIII fibroblasts which suggest that, as expected, the lysosomal-mitochondrial axis is implicated in GHF-201 mechanism of action in GSDIII models as well, with possible implications for amelioration of dysfunctional metabolic pathways reliant on mitochondrial function.

Figures 2 and 3 were modified based on the comments of reviewer 1. We believe that now they are clearer.  

Comment: The last part of the conclusions seems to be too dismissive and you link GSD II without any other explanation. GSD III and II are quiet different on the pathogenesis. Autophagy is common with many other lysosomal disorders. Could you explain better the liver results with this intramuscular treatment and your explanation.

Reply: If the author refers to the word “GSDII” now found in line 411, this was of course a mistake (the original text was “GDE deficiency, or GSDII”, so we meant GSDIII, which is GDE deficiency) . It is now corrected to GSDIII. Thank you for this comment.

If the reviewer refers to the comparison between GSDII and GSDIII in the 3rd paragraph of the discussion, we indeed discuss there the differences between the two diseases, stressing that both disorders manifest autophagic deficiencies. As the reviewer claims, indeed autophagic deficiency is common to most lysosomal storage disorders. What is interesting is that, in common with GSDII, GSDIII also presents with autophagic deficiency even though the deficient enzyme in the disease, GDE, is cytoplasmic.  

The last comment of the reviewer refers to the amelioration of liver dysfunction by an intramuscularly injected compound. Intramuscularly injected substances can be biodistributed to the liver and systemically and intramuscular injection is used in several species for that purpose. For instance, please see Greig et al (2014) PLoS One 9: e112268; Koshkina et al (1999) Cancer Chemother Pharmacol 44: 187 and more.

Reviewer 2 Report

Comments and Suggestions for Authors

Dear Authors,

congratulation for the great quality in the research of this disorder. The biochemical pathways and metabolic views are explained in this paper with great details. At line 98 you talk about combination therapy. I suggest to explain the current treatment in this form og glycogen storage disorder and the clinical trial with gene therapy and mRNA to face the clinical impact of this new therapy you propose.

The comparison between studies on mice and in fibroblasts of humans are very interesting and  you explain very enthusiastic results. We understand that the GSD mouse model is very fragile and it is very complicated to manage.  Did you perform also metabolomic studies or The hipothesis of the involvement of lysosomal-mitochondrial axis also in GSDIII is very challenge to understand. The figures 5 and 6 are very clear, less the figure 2 and 3.

The last part of the conclusions seems to be too dismissive and you link GSD II without any other explanation. GSD III and II are quiet different on the pathogenesis. Autophagy is common with many other lysosomal disorders. Could you explain better the liver results with this intramuscular treatment and your explanation.

Comments on the Quality of English Language

Good quality, only minor editing will be required.

Author Response

(The authors gave the same response as above.)

Round 2

Reviewer 1 Report

Comments and Suggestions for Authors

The article improved significantly after the recent revision, and the authors provided answers to all concerns. While most responses and changes of the manuscript are adequate, some left open further questions, which need to be addressed before publishing. Please find this reviewer’s responses to the authors’ response below in bold and in the attached PDF.

ORIG. REVIEWER: The authors investigate the ability of an autophagy enhancing drug, GHF201, to ameliorate the effects of glycogen debranching enzyme deficiency in mouse models and patient fibroblasts. GHF-201 was previously shown to improve a GSD4 mouse model (Kakhlon et al. 2021). Here, seems to partially improve some behavioral and biochemical parameters affected in Agl-/- mice. The present study as well as the previous study would benefit from showing how the drug impacts all measured parameters in WT. This would allow to distinguish between more general systemic effects of the drug (especially behavioral effects) and those specifically resulting from increased autophagic flux.

AUTHOR RESPONSE: Thank you for this comment. The measured parameters related to accumulation of the pathogenic phosphorylase-limit dextrin (i.e., ultrastructural changes of sarcomeres (Figure 5), and glycogen accumulation (Figure 6A)) will probably not show any response to GHF-201 in WT animals since the latter do not manifest any accumulation of phosphorylase-limit dextrin, or the glycogen vacuoles and glycogen surcharge associated with it. As for other parameters, the effect of GHF-201 on motor/behavioral parameters (importantly gait, but also positional passivity, touch escape, body position, and transfer behavior) and reflexes (extension reflex, righting reflex, trunk curl, tail elevation, pelvic elevation) in WT congenic mice at different concentrations and exposure times has already been tested and published as part of the Irwin tests (Kakhlon et al (2021) EMBO Mol Med 13:e14554). We have also showed in that paper that GHF-201, referred to as 144DG11 in that publication because it was prior to its sublicensing to the GHF company, does not significantly modify weight, weight gain over time and histopathology in these WT C57BL/6J mice, also congenic to the Agl-/- mice (Figure EV2 there). In addition, necroscopic macroscopy performed by an external contractor (Envigo) as part of GHF-201 development for clinical use, showed it has no adverse effect at an organ level. These data cannot be disclosed due to contractual discretion with GHF. Taken together, we believe that these comprehensive examinations show that GHF-201 does not have general systemic effects.

REVIEWER: A systemic effect of GHF-201 on WT animals has only been observed for 1h and 24h. It remains unclear, whether the moderate behavioral differences after months of treatment are stemming from side effects of the drug (e.g. appetite stimulation), rather than from its ability to increase autophagic flux. The author’s should include this limitation of the study in the discussion.

ORIG. REVIEWER: It is apparent that treated Agl-/- mice consume more food, gain more weight, and have more grip strength. Several other parameters that were affected by the drug could be simply consequences of increased food intake. Several parameters are only very slightly changed after treatment. For instance, the in vivo metabolic profiles show very small differences, and insufficient description of methodological detail does not exclude that statistical significance maybe reached due to the use of inappropriate ways of analyses. Similar issues exist with the analyses and interpretation of the fibroblast high content data. The quantification of muscle glycogen by TEM seems unusual. Especially, the reduction of glycogen content needs corroboration as it is the only indication that GHF201 acts truly to mitigate the core problem that exists in GSDIII, namely the inability to degrade phosphorylase-limit glycogen. Overall, the paper does not fully convince this reviewer that GHF201 helps degrade phosphorylase-limit glycogen to an extent that it would be therapeutically relevant.  

AUTHOR RESPONSE: GHF-201 treatment did not cause Agl-/- mice to gain more weight (see for instance Figure 1). While GHF-201 caused a non-statistically significant trend of increased food consumption in the dark in Agl-/- mice (Figure 4C), we would combine this increased food intake with the increased fuel burning and the food requiring energy expenditure associated with the GHF-201 treatment. Put together, we would interpret these observations as an increase in food metabolization efficacy, caused by GHF-201-mediated increased autophagic catabolism. Therefore, we would maintain that food intake does not stand on its own, but should rather be interpreted in conjunction with other metabolic parameters. Such a consolidated interpretation would connect the increased food intake with the increased fuel burning required for the increased energy expenditure.

REVIEWER: I agree that since no weight is gained, the treatment-mediated increased food intake is likely compensated by drug-induced increase in food metabolization. The authors should explain in the discussion why this could not also lead to improved behavioral performance (such as that of grip strength) if WT animals are treated with the drug for a longer period of time, and why the authors assume it has to do with the removal of phosphorylase limit glycogen. Why is the alternative hypothesis that GHF201 simply stimulates appetite, the mice eat more and feel better, are more active (and burn more fat because that’s still easier for them because they are not used to burning carbs)? Please discuss.

Major concerns:

ORIG. REVIEWER: Line 134: please explain in detail by which statistical methods outliers were identified.

AUTHOR REPSONSE: Outliers were removed using the Prism (version 10.2.3) ROUT method with Q=10%. This information was added to the text.

REVIEWER: Agreed.

ORIG. REVIEWER: Line 101: please explain whether WT controls and Agl-/- mice were littermates, i.e. derived from het/het breedings or from independent breedings of Agl-/- with Agl-/- for the generation of KOs and WT with WT for the generation of WT controls.

AUTHOR REPSONSE: WT controls were Agl+/+ littermates derived from heterozygous/heterozygous breeding which also produced the Agl-/- KO mice used in this study and Agl+/- heterozygotes. We added this information to the main text.

REVIEWER: Agreed.

ORIG. REVIEWER: Line 154: Please state whether the observed weight difference is statistically significant. Is it possible that weight differences are caused by non-littermate comparisons? See comment for line 101. If so, please include this as a possible explanation in section 3.1.

AUTHOR REPSONSE: Yes the difference in weight over time between WT and KO mice is statistically significant. This information was added to the main text. As stated, WT mice were Agl+/+ littermates derived from heterozygous/heterozygous breeding which also produced the Agl-/- KO mice used in this study.

REVIEWER: Agreed.

ORIG. REVIEWER: Figure 1: In this figure weight has been determined every 6-7 days. This is inconsistent with the description in the method part, which indicates biweekly weight monitoring. Please consolidate. Please include information on the number of mice used in each average, and whether (and when and from which group) any mice were removed due to >20% weight loss. In addition the weight at treatment initiation is not recorded, precluding the authors to evaluate weight in comparison to treatment initiation (this is one criterion for trial exclusion mentioned in the method section line 106). The authors should clarify this in the text.

AUTHOR REPSONSE: According to the IACUC approval of this protocol, animal weights have to be recorded biweekly to monitor weight deviations which can indicate health deterioration. This is the reason for indicating biweekly weight recording in the Methods section. In practice, we increased the weighting frequency to once a week so that it can be reported as a result and so that we will have enough data points for statistical analysis. We corrected the weighting frequency in the Methods section.

Mouse numbers were added to the legend of Figure 1.

We added the number, times and groups of mouse removal due to excessive weight loss (it was actually >10% in two consecutive biweekly measurements, rather than >20% from treatment initiation).

We have measured the locomotor parameters only at the indicated times after treatment initiation. Our goal was to compare Agl-/- treated with Agl-/- untreated mice using WT mice as a reference. To that end we believe that comparisons made at fixed times after treatment initiation are sufficient, as we are not comparing change from baseline, but rather the change, or KO response to treatment, itself. The fact that this change is compared at the same time points after treatment initiation can exclude time as a factor influencing the response to the treatment. This experimental setup, where movement in an open field is measured at fixed time points after treatment initiation (and not at initiation itself) is also used in several high impact publications, e.g., Zhai et al (2023) Gene Therapy 30: 75; van Buel et al (2015) J Neuroinfl 15: 232; Li et al (2018) Biol Reproduction 99: 349; and also our own publication on GSD type IV Kakhlon et al (2021) EMBO Mol Med 13:e14554.

REVIEWER: Agreed.

ORIG. REVIEWER: Figure 2: Please include data at study initiation.

AUTHOR REPSONSE: Weight at initiation of treatment was added.

REVIEWER: Agreed.

ORIG. REVIEWER: Figure 2D: This figure is hard to interpret, especially due to the stacking of the data series. Also, there are no statistical analyses indicated that support the authors’ claim that Wild type mice manifest training and motor learning capacity, but KO mice do not. Please use a different representation of the data, where (1) latencies are not stacked, (2) consecutive trials are next to each other and grouped by day post treatment, (3) error bars remain included, and (4) statistical test results are added. The drawing below may provide a helpful starting point (drawing only included in the submitted review PDF).

AUTHOR REPSONSE: We have corrected the figure as per the reviewer suggestion and added statistical analyses which indeed added more details about the differences between the different groups. We thank the reviewer for these suggestions.

REVIEWER: The figure has greatly improved. The authors should indicate in the text, how they define ‘training capacity’ and ‘motor learning capacity’. Also they should indicate which averages were compared to assess the training capacity.

ORIG. REVIEWER: Line 173-176: The authors mention a weight drop of all mice by >10% from the previous weight reading. This is inconsistent with weight recordings in Fig.1, where no average weight drop of >10% around post treatment day 100 can be observed.

AUTHOR REPSONSE: We guess the reviewer refers to lines 181-183 because weight drop of mice is not mentioned in lines 173-176. We mention that the weight of all mice was decreased by >10% after the 4th test. This decrease is indeed not shown in Figure 1 because by the time it took place at day 154 after treatment initiation (N.B., not around day 100, the previous plot continued until day 138), we have already discontinued the experiment and were not able to perform the 5th test. Originally, the weight over time plot was meant to encompass tests times. We have now added these additional two time points even though the 5th test did not take place.

REVIEWER: How do the authors explain the sharp WT drop even in vehicle-treated WT animals? The authors should add observation and explanation off the weight-drop in the weight paragraph.

ORIG. REVIEWER: Fig. 3: please include individual data points for each bar. Given that more than 2 groups are compared, simple T-tests are not appropriate. Please conduct ANOVA with appropriate post-hoc tests and multiple comparison adjustment of p-values. Indicate which test is used and state the significance threshold.

AUTHOR REPSONSE: We included data from individual animals in each bar. Statistical significance analyses (comparisons between GHF-201-treated and untreated Agl-/- mice) were now conducted by Two Way ANOVA with repeated measures with Tukey post-hoc test using simple effects within rows (times) analysis. This is indicated in the revised legend of Figure 3.

REVIEWER: Agreed.

ORIG. REVIEWER: Fig.4: The organization of the figure is too rough. Please substructure the figure by assigning more panel letters. Calculation of the averages in the bar charts is not explained. Please show individual data points. Those should be 4 in each bar, as n=4. The time points should first be averaged per mouse. Then averages should be calculated across the four biological replicates. Statistical tests used in the bar charts are not explained. They should be ANOVA with appropriate post-hoc. Significance threshold must be declared.

AUTHOR REPSONSE: The figure was subdivided to more panels as requested. Averaging the time points per mouse and comparing them by normal ANOVA could be problematic because time itself also affects the results or contributes to variability (p<0.1, Two Way ANOVA analysis performed to demonstrate this point (i.e., not a part of the manuscript)). Therefore, we performed instead of ANOVA, ANOVA with repeated measures to account for the effect of time and compare the different courses of change in the means, rather than the means themselves. This approach for the analyzing indirect calorimetry in metabolic cages was also taken by other workers, such as John et al. (2022) Commun Biol 5: 946 and Adriaenssens et al. (2023) JCI Insight 8: e164921. We also show the results as differences in means, which are the output of repeated measures ANOVA, between WT/Veh vs KO/Veh to show the disease effect, WT/Veh vs KO/GHF-201 to show whether GHF201 treatment brings the parameter tested closer to WT levels, suggesting amelioration, and KO/Veh vs KO/GHF-201 to test treatment effect. According to the mouse numbers, n=4 mean differences are shown for each bar in the bar graphs summarizing the longitudinal data. This new statistical analysis changed some of the results compared to the previous statistical analysis, which was based on pooling all data points in a group. However, the improvement of all metabolic parameters by GHF-201 according to the previous analysis is sustained in this new analysis. Analysis was performed by One Way ANOVA (to analyze the entire time-dependent trend, rather than each separate time point in Two Way ANOVA) with repeated measures and Tukey post-hoc test and p values are now indicated.

REVIEWER: The panel description of the figure improved. However, the statistical methods should be explained in much more detail in the method section. Moreover, it is still not clear how the bar charts in A1,3,5 etc. are derived from the time graphs. The calculations should be explained in the method section. Regarding the results, the authors should take more care in explaining the results in the results part. For EACH parameter, it would be advisable if the explanation starts with a statement about how the KO/Veh differs from WT. This should be followed by how the treatment changes the parameter in relation to KO/Veh and in relation to WT. It should be pointed out in the text that for some parameters there are only a small difference between WT and KO, which only become larger upon treatment. Examples are TEE light, Fat Oxidation light, wheel running light. Here the authors should not speak of a rescue. For consistency, the authors should also split the food and water intake into light and dark phase. They should discuss the effect of the treatment on food intake and how this could lead to improvement of the behavioral phenotypes. Please explain in the text why food and water intake are not similar at the end of the 24h period and at the beginning (shouldn’t intake decline toward the end of the dark period?)?

ORIG. REVIEWER: Re In vivo metabolic profile: Please include detailed descriptions of the XYZ beam and wheel running assays used. The information in the referenced publication by Kakhlon et al. 2021 (EMBO Mol Med) is not sufficient to understand the method. The authors should discuss how GHF201 can lead to increased ambulatory activity in the light and decreased activity in the dark. In addition the authors should explain why the same drug leads to increased ambulatory activity in the dark in GBE1-deficient mice (Kakhlon et al. 2021).

AUTHOR REPSONSE: Thank you for the comment. Explanations on the XYZ beam and wheel running systems was added in section 2.1.

As for the light-selective ameliorative effect of the GHF201 effect, this is explained in the one before last paragraph of the Discussion: “Interestingly, GHF-201-mediated improvement of fat oxidation (Figure 4A), and other in vivo metabolic effects (TEE, (Figure 4A), ambulatory activity (Figure 4B), wheel running (Figure 2B)) were more pronounced in the light than in the dark. Since mice are nocturnal animals, in which food is mostly consumed in the dark (see also Figure 4C) and digested in the light, it can be conjectured that GHF-201, affecting fuel digestion, or autophagic substrate catabolism, exerts a larger effect in the light.”

Following the reviewer comment, we also added the following text there: “Along these lines, a possible explanation for the GHF201-associated decrease of activity in the dark is that the untimely stimulation of fuel-generating catabolism in the dark (Figure 4B1) might lead to fuel excess known to be detrimental as it might lead to mitochondrial fragmentation in pancreatic beta cells (e.g., Alan & Scorrano (2022) Current Biology 32, R589–R683) and unbalanced reactive oxygen species production (Fisher-Wellman et al (2013) Free Radic Biol Med 65:1201).”

Regarding the fact that GHF-201 increased ambulatory activity in the dark in GSDIV modeling mice, while decreasing it in GSDIII modeling mice, this difference can be explained in several ways. However, one important difference between these two diseases is that in GSDIV patients, and in GSDIV-modeling mice, polyglucosan bodies (PB) formed by brancher deficiency are architecturally different than phosphorylase-limit dextrin in GSDIII formed by debrancher deficiency. The large size and unique amylopectin architecture of PB in GSDIV models causes them to clog axons, which is not the case for the aberrant glycogen (phosphorylase-limit dextrin) in GSDIII. Therefore, PB-clearance from axons and peripheral nerve by GHF201 in a GSDIV mouse model (Kakhlon et al (2021) EMBO Mol Med 13:e14554) might enhance mobility by improving nerve-muscle communication. This clearance of the noxious PB from axons and neuromuscular junctions might override the putative detrimental effect of excess fuel in the dark and so might be more important for locomotion and enhance it also in the dark in GSDIV mice, as opposed to GSDIII mice.

REVIEWER: Agreed.

ORIG. REVIEWER: Line 215-221: It is questionable whether EM is the best method to quantify muscle glycogen accumulation and rescue thereof in mouse muscle. EM provides only very small fields of view and, to avoid conclusions drawn from non-representative sections and images, several sections and images per mouse should be analyzed. Please explain (in the method part) how many sections from how many animals were analyzed. Quantified areas from different sections of the same animal should be averaged first. Each animal should be represented by only one data point in Fig. 5B.

AUTHOR REPSONSE: EM has already been used to quantify muscle glycogen in GSIII mouse models and patients (Laforet et al (2019) Acta Neuropathol Commun 7: 167; Vidal et al (2018) Mol Ther 26: 890). This method in muscle is also described in a JoVE article (https://app.jove.com/v/63347/quantification-of-subcellular-glycogen-distribution-in-skeletal-muscle-fibers-using-transmission-electron-microscopy). The quantification described here is based on 10 sections per muscle, which are considered representative of a muscle fiber which they sample across its entire area (https://app.jove.com/v/63347/quantification-of-subcellular-glycogen-distribution-in-skeletal-muscle-fibers-using-transmission-electron-microscopy).

REVIEWER: Agreed.

ORIG. REVIEWER: The EM image shown for the non-treated GSD3 mouse looks very different in format section plane and overall contrast from the other two, which makes this reviewer question whether the authors selected comparable sections. The authors should include additional images from different mice in the supplement. Furthermore, the authors should corroborate their EM data by biochemical quantification of glycogen.

AUTHOR REPSONSE: All TEM images were acquired at the same magnification, acceleration voltage, and emission amperage. Therefore, all images are comparable. The reason for the different appearance of GSD3 mouse muscle images is biological and reflects disruption of the muscle fibers by the glycogen granules. A similar visual effect of the GSD3 pathology in TEM images of muscle fibers was observed in GSD3 patients (Laforet et al (2019) Acta Neuropathol Commun 7: 167) and murine muscle (Vidal et al (2018) Mol Ther 26: 890). As requested, additional figures were added to a supplemental figure.

We accept the reviewer’s critique that glycogen in the muscle fibers should also be quantified biochemically. We added data of total glycogen levels in muscle samples from the same animals analyzed by TEM experiments. We also updated the Methods section accordingly. These results show the same trend as the TEM results and thus support them.

REVIEWER: Please add details to the method section explaining the glycogen extraction method chosen for biochemical quantification in muscle. Which buffer was used? Which controls have been used in the assay? Etc.

ORIG. REVIEWER: Fig. 6: Please add panel letters. As analyses were conducted with n=3, averages should be calculated across the three biological replicates, not across all individual images taken. This dictates that bars should only contain 3 data points each, representing the level of biological replication. Statistical significance should be calculated on the basis of n=3 after averaging of images taken from the same biological replicate. The method section related to this figure mentions that outliers were removed. The authors should describe the method they used to identify outliers. Please also describe the statistical tests used.

AUTHOR REPSONSE: Thank you for these comments. Panel letters were added. Data from each subject (patients and healthy controls) are now presented separately and statistical significance was recalculated by Two Way ANOVA with variability among images from each subject as one of the factors and intergroup variability as the other. This information was added to the figure legend. As mentioned above, as a reply to another question from this reviewer, outliers were removed using the Prism (version 10.2.3) ROUT method with Q=10%. This information was added to the Methods section under statistical analysis.

REVIEWER: Agreed.

Minor concerns:

REVIEWER: all minor concerns have been appropriately addressed.

Author Response

The article improved significantly after the recent revision, and the authors provided answers to all concerns. While most responses and changes of the manuscript are adequate, some left open further questions, which need to be addressed before publishing. Please find this reviewer’s responses to the authors’ response below in bold and in the attached PDF.

We share the reviewer’s opinion that the manuscript has improved by addressing the concerns the reviewer brought. We thank the reviewer for that.

Comment 1: A systemic effect of GHF-201 on WT animals has only been observed for 1h and 24h. It remains unclear, whether the moderate behavioral differences after months of treatment are stemming from side effects of the drug (e.g. appetite stimulation), rather than from its ability to increase autophagic flux. The author’s should include this limitation of the study in the discussion.

Response 1: Irwin test (absence of) effects were indeed obtained after 1h and 24 h exposures to GHF201, as per the common use of Irwin tests. However, the effects we mentioned in our previous response (or rather their lack of) on weight and histopathology are actually long-term (3 months) effects. In Figure EV2 in our paper Kakhlon et al. (2021) EMBO Mol Med 13: e14554, we show no significant effect of GHF201 on body weight over time (Figure EV2A) and on histology (Figure EV2B). These results were obtained after a long-term 3 months exposure of WT animals to GHF201.We believe that since no significant weight change were observed following this long-term exposure of the animals to GHF201, its effects on Agl-/- mice behavior reported here are not related to appetite stimulation, or any other general physiological changes not related to autophagy enhancement. We briefly alluded to that in the Discussion (end of 3rd paragraph), as per the reviewer suggestion.

Comment 2: I agree that since no weight is gained, the treatment-mediated increased food intake is likely compensated by drug-induced increase in food metabolization. The authors should explain in the discussion why this could not also lead to improved behavioral performance (such as that of grip strength) if WT animals are treated with the drug for a longer period of time, and why the authors assume it has to do with the removal of phosphorylase limit glycogen. Why is the alternative hypothesis that GHF201 simply stimulates appetite, the mice eat more and feel better, are more active (and burn more fat because that’s still easier for them because they are not used to burning carbs)? Please discuss.

Response 2: As explained in the previous response, WT animals were indeed treated for a long-term period (3 months). We assume that the beneficial effect of GHF201 is specific to Agl-/- mice because the accumulation of phosphorylase-limit dextrin (PLD) in this strain disrupts muscle structure as can be observed in Figure 5 and in the new supplemental figure. Therefore, partial removal of the pathogenic PLD in Agl-/- mice can more noticeably relieve the motor dysfunction caused by the disrupted muscle structure than in WT mice where such a disruption and consequent motor dysfunction do not exist. In other words, we believe that the beneficial effect of GHF201is more discernible in affected animals where there is more room for phenotype improvement. We believe that, as explained, GHF201 is more efficacious in animals predisposed to damage associated with buildup of noxious substrate (e.g.,  PLD in Agl-/- mice). However, since autophagy is generally considered a cell protective process, it might improve animal health in more than one way, even in WT mice. While we did not test here the effect of GHF201 on the motor function of WT mice, we can envision that enhancement of catabolic autophagy by GHF201 might exert a general positive effect on body energy and thus improve various health aspects in both WT and transgenic mice. This information was also added to Discussion at the 3rd paragraph.

Comment 3: The figure has greatly improved. The authors should indicate in the text, how they define ‘training capacity’ and ‘motor learning capacity’. Also they should indicate which averages were compared to assess the training capacity.

Response 3: Training and motor learning capacities are defined in lines 192-193: Training and motor learning capacity were, respectively, demonstrated by increased latency between runs in each test and by increased overall latency over time.” We explained which averages were compared to assess the differences in training capacities (average run latencies at each time post treatment) in lines 195-198.

Comment 4: How do the authors explain the sharp WT drop even in vehicle-treated WT animals? The authors should add observation and explanation off the weight-drop in the weight paragraph.

Response 4: The weight drop in the last two time points is unintentional and not related to the treatment, or to transgenic pathology as it also occurred in wild type animals (as the reviewer pointed out). Our animal facility is AAALAC-accredited and observes a strict policy of monitoring temperature, humidity, light cycle, food and water supply, and pathogens. We checked and no significant deviations in these parameters were recorded during the period in which weight was reduced. Thus, the two possible explanations we can offer for this weight drop are: 1. They reflect a normal fluctuation, which crossed the 10% weight-drop inter-weighting threshold. This 10% threshold is an average value, based on experience. However, there is a small chance that mice whose weight drops by 10 % or more between two biweekly weightings could also recover. This might have been the case in the last two time points since we did not observe any conspicuous health deterioration in the mice such as poor grooming, wounds etc. Therefore, had we continued the experiments, mice could have recovered the weight loss, but we had to euthanize them according to our IACUC policy. 2. While animals in our facility are closely monitored by expert veterinarians and supervised and audited by the Israel Ministry of Health and AAALAC, errors can always happen, even though at a low chance. Error in recording data or in monitoring might have occurred for the last two time points, especially since the same trend (weight reduction) occurred in all cages.

Usually, animal weight during treatment is monitored and reported only in parallel to reporting the effect of the treatment on other parameters because the purpose is to possibly correlate the effect of the treatment on these parameters and on animal weight. This is practiced in numerous works, e.g., Ma et al. (2009) Anticancer Res 29: 2421; Nestor et al. (2022) Sci Rep 12: 6666; and many more including our own work Kakhlon et al. (2021) EMBO Mol Med 13: e14554. Since the weight drop leading to the discontinuation of the experiment took place before the next (5th) behavioral/motor test should have taken place, we returned the original Figure 1, which does not include weight recording outside of test times. We added a statement that the tests and animal weightings were discontinued due to an unexplained >10% weight drop between two weightings in all animals.

Comment 5: The panel description of the figure improved. However, the statistical methods should be explained in much more detail in the method section. Moreover, it is still not clear how the bar charts in A1,3,5 etc. are derived from the time graphs. The calculations should be explained in the method section. Regarding the results, the authors should take more care in explaining the results in the results part. For EACH parameter, it would be advisable if the explanation starts with a statement about how the KO/Veh differs from WT. This should be followed by how the treatment changes the parameter in relation to KO/Veh and in relation to WT. It should be pointed out in the text that for some parameters there are only a small difference between WT and KO, which only become larger upon treatment. Examples are TEE light, Fat Oxidation light, wheel running light. Here the authors should not speak of a rescue. For consistency, the authors should also split the food and water intake into light and dark phase. They should discuss the effect of the treatment on food intake and how this could lead to improvement of the behavioral phenotypes. Please explain in the text why food and water intake are not similar at the end of the 24h period and at the beginning (shouldn’t intake decline toward the end of the dark period?)?

Response 5: Statistical methods of the metabolic cages analysis are now detailed in lines 164-175 in the Methods section under “Statistical Analysis.” As requested, we now also explain in this section how the bar charts of mean differences were derived from the time-resolved plots to their left.

In the results section we now explain for each metabolic parameter (analyzed in Figure 4) how transgenic Agl-/-  mice differ from WT mice, how GHF-201 affects transgenic mice and how the effect of GHF201 on transgenic mice modifies the similarity of the treated transgenic to WT mice.

Regarding the differences between WT and KO, besides the extent or magnitude of the difference, what matters is its direction: Whether it improves (enhances) the metabolic parameter in KO mice so that its levels get closer to those of WT mice (this is called a rescue effect assuming WT levels are healthier), or whether the treatment-caused enhancement supersedes WT levels of the parameter (an “overshoot” effect). In some cases, such as fat oxidation in our case (Figure 4A5), KO levels are even higher than those of WT. As the reviewer mentions, the parameters for which there is a small difference between WT and KO which is enhanced by the treatment (i.e., where the mean difference between WT and KO/GHF201 are larger than the difference between WT and KO) are: non-weight-adjusted TEE in the light (Figure 4A3), where GHF-201 created an overshoot effect, fat oxidation in the light (Figure 4A5), where GHF201 further enhanced the levels already enhanced in KO vs WT, and wheel running in the light (Figure 4B2), where too GHF201 had an overshoot effect. For weight-adjusted TEE in the light and overall (Figures 4A2 and 4A4), there is actually a rescue effect, indeed not found in non-weight adjusted animals, because following the treatment, non weight-adjusted TEE in the light and overall periods became indistinguishable from WT. All these trends were emphasized in the text in section 3.4 in Results.

Food and water intake were split into light, dark and overall periods. The improvement in food intake was negligible. However, the improvement in water intake, which usually accompanies increased activity, is now mentioned in that context. Since both food and water intake are cumulative graphs, they should increase with time, as they indeed do, and not become similar to the initial values towards the end of the 24 h period. This is now explained in the text at the end of section 3.4.

Comment 6: Please add details to the method section explaining the glycogen extraction method chosen for biochemical quantification in muscle. Which buffer was used? Which controls have been used in the assay? Etc.

Response 6: This method was added in the Methods lines 136-143.